# Flattop regulates basal body docking and positioning in mono- and multiciliated cells

**Moritz Gegg[1,2], Anika Böttcher[1,2], Ingo Burtscher[1,2], Stefan Hasenoeder[1,2], Claude Van Campenhout[3], Michaela Aichler[4], Axel Walch[4], Seth G N Grant[5,6], Heiko Lickert[1,2]***

[1]Institute of Stem Cell Research, Helmholtz Center Munich, Munich, Germany; [2]Institute of Diabetes and Regeneration Research, Helmholtz Center Munich, Munich, Germany; [3]Genetique du Developpement, L'Institut de biologie et de médecine moléculaires, Université libre de Bruxelles, Gosselies, Belgium; [4]Research Unit Analytical Pathology, Helmholtz Center Munich, Munich, Germany; [5]Centre for Clinical Brain Sciences, University of Edinburgh, Edinburgh, United Kingdom; [6]Centre for Neuroregeneration, Univeristy of Edinburgh, Cambridge, United Kingdom

**Abstract** Planar cell polarity (PCP) regulates basal body (BB) docking and positioning during cilia formation, but the underlying mechanisms remain elusive. In this study, we investigate the uncharacterized gene *Flattop (Fltp)* that is transcriptionally activated during PCP acquisition in ciliated tissues. *Fltp* knock-out mice show BB docking and ciliogenesis defects in multiciliated lung cells. Furthermore, Fltp is necessary for kinocilium positioning in monociliated inner ear hair cells. In these cells, the core PCP molecule Dishevelled 2, the BB/spindle positioning protein Dlg3, and Fltp localize directly adjacent to the apical plasma membrane, physically interact and surround the BB at the interface of the microtubule and actin cytoskeleton. Dlg3 and Fltp knock-outs suggest that both cooperatively translate PCP cues for BB positioning in the inner ear. Taken together, the identification of novel BB/spindle positioning components as potential mediators of PCP signaling might have broader implications for other cell types, ciliary disease, and asymmetric cell division.

***For correspondence:** heiko. lickert@helmholtz-muenchen.de

**Competing interests:** The authors declare that no competing interests exist.

**Reviewing editor**: Jeremy Nathans, Howard Hughes Medical Institute, Johns Hopkins University School of Medicine, United States

## Introduction

The conserved PCP signaling pathway regulates the orientation of cells and organelles within the plane of an epithelium and is crucially important for developmental patterning as well as organ morphogenesis, homeostasis, and physiology (*Seifert and Mlodzik, 2007*; *Wang and Nathans, 2007*; *Peng and Axelrod, 2012*; *Wallingford, 2012*). Pioneering studies in *Drosophila* and *Xenopus* have revealed that global, non-cell autonomous, and cell intrinsic signaling mechanisms act in concert to establish tissue polarity. Core PCP molecules including Van Gogh-like (Vangl1-2), Cadherin EGF LAG seven-pass G-type receptor (Celsr1-3), Frizzled (Fzd3, 6), Dishevelled (Dvl1-3), and Prickle (Pk1-2) are localized asymmetrically at the cell cortex to provide polarity information for morphogenesis and oriented cell division. Significant progress has been made in understanding the asymmetric core PCP localization in vertebrates but it is less clear how this regulates cytoskeletal rearrangements that drive morphogenesis via tissue specific downstream effector molecules (*Wallingford, 2012*). Thus, the identification of novel PCP effectors that indicate pathway activity and mediate signaling and/or morphogenesis will be the key to unravel the function of this molecular pathway in development and disease.

Besides the Rho family of GTPases, which are also implicated in apical–basal (A–B) polarity establishment, the best-studied PCP effector molecules are Inturned (Intu) and Fuzzy (Fuz)

**eLife digest** Epithelial tissues are sheets of cells that line the surface of many parts of the body, including the airways and the inner ear. Small hair-like structures called cilia can be found on the top surface of many epithelial cells and are arranged in a precise, ordered pattern. Such patterning ensures that cilia can work in a co-ordinated manner, for example by beating together to help clearing mucus from airways.

Cilia grow out from 'basal bodies' and, like many other important structures in a cell, these basal bodies must be oriented along the correct side of an epithelial tissue. This is achieved by 'planar cell polarity signaling', which makes sure that the structures inside a cell are correctly aligned, and ensures that polarized cells themselves are correctly oriented across the epithelial tissue. Disruption of this signaling can result in developmental defects.

Some proteins help to establish polarity in a cell by altering the cell's cytoskeleton—the structural support and transport network of the cell. A 'core' complex of proteins then coordinates how the cells are arranged throughout the epithelial tissue. Although many of the proteins involved in each of these roles are known, how they interact with each other to establish planar cell polarity remains poorly understood.

Now, Gegg et al. report that, in mice, a protein called Flattop functions to position basal bodies—and thus cilia—by working together with another protein called Dlg3. In mice that cannot produce Flattop, cilia formation is defective in the lung, and the cilia in the inner ear are positioned incorrectly. Gegg et al. found that in the inner ear, Flattop and Dlg3 physically interact with each other and two other proteins—including one of the core proteins involved in planar cell polarity. This protein complex then surrounds the basal bodies at the point where they connect to the cell's cytoskeleton.

Future challenges will be to clarify how the protein complex anchors to the cytoskeleton and how it interacts with other core planar cell polarity proteins in the cells of the inner ear. It will also be important to see whether this protein complex fulfills a similar role in other ciliated epithelial tissues.

---

(*Collier and Gubb, 1997*; *Park et al., 2006*, *2008*; *Gray et al., 2009*). Both directly regulate ciliogenesis by mediating the assembly of the apical actin cytoskeleton but are not required for the polarized accumulation of core PCP components. The core PCP molecule Dvl2 localizes near the base of cilia and functions together with Intu and Rho GTPases to dock and polarize BBs for cilia formation and directed ciliary beating (*Park et al., 2008*). BBs are amplified deep in the cytoplasm of multiciliated cells (MCCs) and apical plasma membrane (PM) transport depends on Dvl and the vesicle trafficking protein Sec8. Up-to-date it is not understood how core PCP molecules physically connect to effector molecules, how this leads to asymmetric membrane polarization and cytoskeletal rearrangements, and if these mechanisms are conserved among different cell types in various organs and during evolution.

First functional evidence for PCP in lung development came from the analysis of Celsr1, Vangl2, and Scribble (Scrib) mutant mice, which showed defects in branching morphogenesis and narrowed lung airways due to cytoskeletal and junctional defects (*Yates et al., 2010*). Multiciliated lung cells first arise at embryonic day (E) 14.0 in the trachea as well as in the main bronchi (*Jain et al., 2010*). Similar to the mucociliary epithelium in frog, differentiation depends on BB amplification, docking, and orientation that allows the formation of hundreds of motile cilia. The differentiation of multiciliated lung cells and the dynamics of the underlying cell biological processes can be modeled in air liquid interface (ALI) cultures of primary mouse tracheal epithelial cells (mTECs) (*You et al., 2002*; *Vladar and Stearns, 2007*; *Vladar et al., 2012*). Asymmetric localization of core PCP molecules at apical junctions regulates the orientation of motile cilia along the longitudinal tissue axis for directed beating and mucus clearing. This likely interdepends on non-cell autonomous cues and intrinsic polarized microtubule (MT) network topology (*Vladar et al., 2012*). Currently, PCP effector molecules that link core molecules, BBs, polarized MTs, and the actin cytoskeleton have not been identified. A better understanding of these molecular processes could provide further insight into a multitude of ciliary dysfunction syndromes of the lung and other organs.

The best-established model to study PCP in vertebrates is the organ of Corti in the inner ear (IE). Mechanosensory hair cells (HCs) are arranged in one inner (IHC) and three outer HC (OHC) rows. The lateral polarization of the V-shaped actin-based stereocilia bundles on HCs strongly depends on ciliogenesis and PCP for proper sound perception (*Montcouquiol et al., 2003*; *Wang et al., 2005*, *2006*; *Jones and Chen, 2008*). Core PCP molecules like Celsr1, Dvl2/3, Fz3/6, and Vangl2 are localized to distinct apical membrane compartments of HCs and supporting cells (*Ezan and Montcouquiol, 2013*). This differential localization seems not sufficient to instruct morphogenesis of actin-rich hair bundles in mammals (*Jones and Chen, 2008*). Instead, it depends on opposing localization of evolutionarily conserved spindle positioning and apical polarity proteins that serve as a blueprint for kinocilium migration and bundle formation (*Ezan et al., 2013*; *Tarchini et al., 2013*). It remains unclear, how core PCP molecules couple to spindle positioning complexes and the actin cytoskeleton to orchestrate morphogenesis.

Spindle positioning proteins as well as the actin and MT cytoskeleton act together with cues from the cell cortex, such as apical junctions and polarity proteins to direct spindle positioning in mammalian epithelial cells (*Kunda and Baum, 2009*). In addition to Inscuteable (mInsc in mammals), Partner of Inscuteable (mPins/LGN in mammals) and the G-protein coupled receptor Gα$_i$ and other evolutionary conserved proteins, such as the basolateral determinants and septate junction localized proteins Dlg and Scrib, regulate A–B spindle positioning (*Knoblich, 2008*; *Siller and Doe, 2008*). In mammals, five *Dlgs* have been identified and we have recently shown that their function has diverged during evolution (*Van Campenhout et al., 2011*). Interestingly, Dlg3 is the only family member that is involved in apical transport and required for tight junction (TJ) consolidation. Taken together, the function of *Drosophila* Dlg in spindle positioning (*Bellaiche et al., 2001*; *Johnston et al., 2009*; *Bergstralh et al., 2013*) and mammalian Dlg3 in PCP establishment (*Van Campenhout et al., 2011*), suggested to us that Dlg3 could mediate PCP-dependent BB positioning.

In this study, we identified *Fltp* as a gene expressed in regions of active PCP signaling including the node, the MCCs of the lung, and the sensory hair cells of the inner ear. Knock-out analysis revealed that Fltp is required for BB docking and cilia formation in the lung as well as BB and kinocilium positioning in the IE. Using ALI cultures, we show that *Fltp* expression is induced while BBs are amplified and docked at the apical PM in differentiating MCCs. Fltp localizes next to BBs, and MT plus ends in the apical actin network and is required for efficient BB docking and cilia formation. We provide evidence that Dlg3 functions together with Fltp to position BBs and kinocilia in the inner ear. Dlg3 and Fltp physically interact with each other, the core PCP protein Dvl2, and the pericentriolar matrix protein γ-Tubulin, suggesting that we have identified a novel BB positioning complex in the inner ear. Together, our data implicate that Fltp is a novel regulator important for BB docking and positioning in mono- and multiciliated cell types that acquire PCP.

## Results

### Identification of *Fltp* as a Foxa2 target gene

We discovered *Fltp* as a functionally non-annotated gene (*1700009P17RiK*) in a screen to identify Foxa2 target genes involved in polarity establishment (*Burtscher and Lickert, 2009*) and expressed in monociliated node cells (*Tamplin et al., 2008*; *Kinzel et al., 2010*). The enrichment of the transcription factor FOXA2 on conserved binding sites in the promoter region of the human *FLTP* ortholog (*C1Orf192*), suggests that *FLTP* is a direct target of FOXA2 (*Figure 1A*) (*Weedon et al., 2014*). The murine *Fltp* gene consists of six exons and the spliced mRNA codes for an open-reading frame (ORF) of 567 nucleotides that translates into a protein of 189 amino acids (*Figure 1B,C*). An alignment of the proteins of different species reveals a high conservation during evolution as well as an N-terminal SH3 binding domain and a C-terminal proline-rich repeat (PRR) (*Figure 1C*). Interestingly, the mouse *1700009P17RiK* cDNA was also identified in a screen for mRNAs highly abundant in ciliated tissues (*McClintock et al., 2008*).

To analyze the Fltp protein in more detail, we raised two polyclonal rabbit antibodies against a central and a C-terminal epitope (*Figure 1C*). We confirmed the specificity of these antibodies in western blot analysis and immunocytochemistry in transiently transfected HEK293T cells (*Figures 1—figure supplement 1A–G, 2D*). Endogenous Fltp protein localizes to the apical PM, the BB, and the primary cilium in monociliated mouse node cells, suggestive for a function of Fltp in BB transport, positioning, and/or ciliogenesis (*Figure 2F,G*).

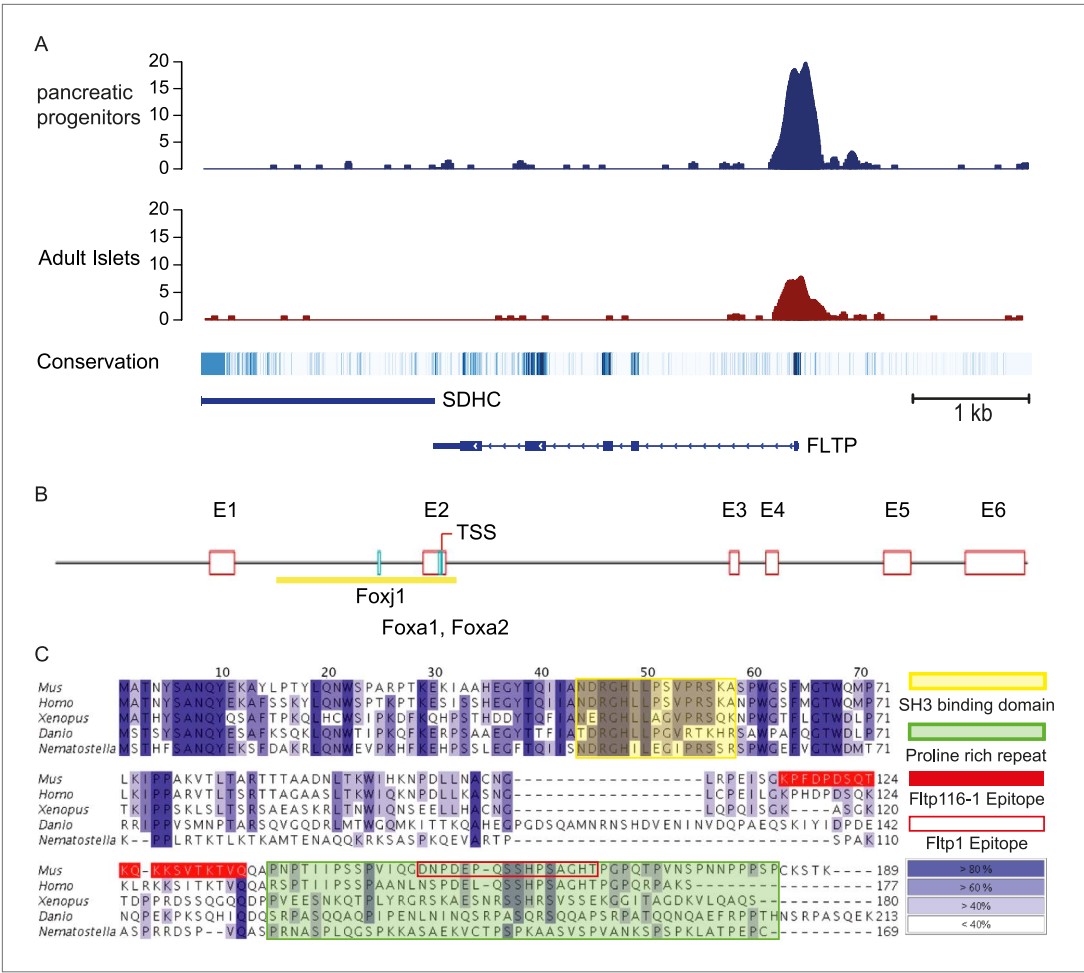

**Figure 1**. *FLTP* has active *FOXA2* binding sites in its promoter and is conserved among species. (**A**) A high amount of the endodermal transcription factor FOXA2 binds the human *FLTP* promoter in pancreatic progenitors and in adult islets, indicating that *FLTP* is a direct target of FOXA2 and expressed in these cells. (**B**) *Fltp* shows predicted (Genomatix) Foxj1, Foxa1, and Foxa2 binding sites in its promoter (clear red boxes: exons (E1–E6); yellow box: promoter; TSS: transcriptional start site; light blue boxes: Foxj1, Foxa1, Foxa2 binding sites). (**C**) Fltp protein alignment shows high conservation between different species (highest conservation in the first 76 amino acids). The mouse and human proteins are highly homologous (yellow box: SH3 binding domain; green box: predicted proline rich repeat (PRR); red filled box: peptide sequence of the Fltp116-1 epitope; red empty box: peptide sequence of the Fltp1 epitope; dark blue indicates conservation over 80%; lighter colors indicate less conservation).

The following source data and figure supplements are available for figure 1:

**Source data 1**. Mendelian ratio of Fltp intercrosses on different backgrounds.

**Figure supplement 1**. Fltp antibody specificity.

**Figure supplement 2**. Fltp construct and confirmation of the knock-out strategy.

## *Fltp* is expressed in mono- and multiciliated tissues during PCP acquisition

To explore *Fltp* expression and function in vivo, we generated an *Fltp^{ZV}* knock-in/knock-out allele by replacing the entire ORF by a multicistronic lac<u>Z</u>-<u>V</u>enus reporter cassette (***Figure 1—figure supplement 2A*** and Supplemental 'Materials and methods'). PCR genotyping as well as Southern and western-ern blotting analysis confirmed the targeted homologous recombination and the generation of a null allele (***Figure 1—figure supplement 2B–D***). *Fltp* homozygous knock-out mice are born at roughly the

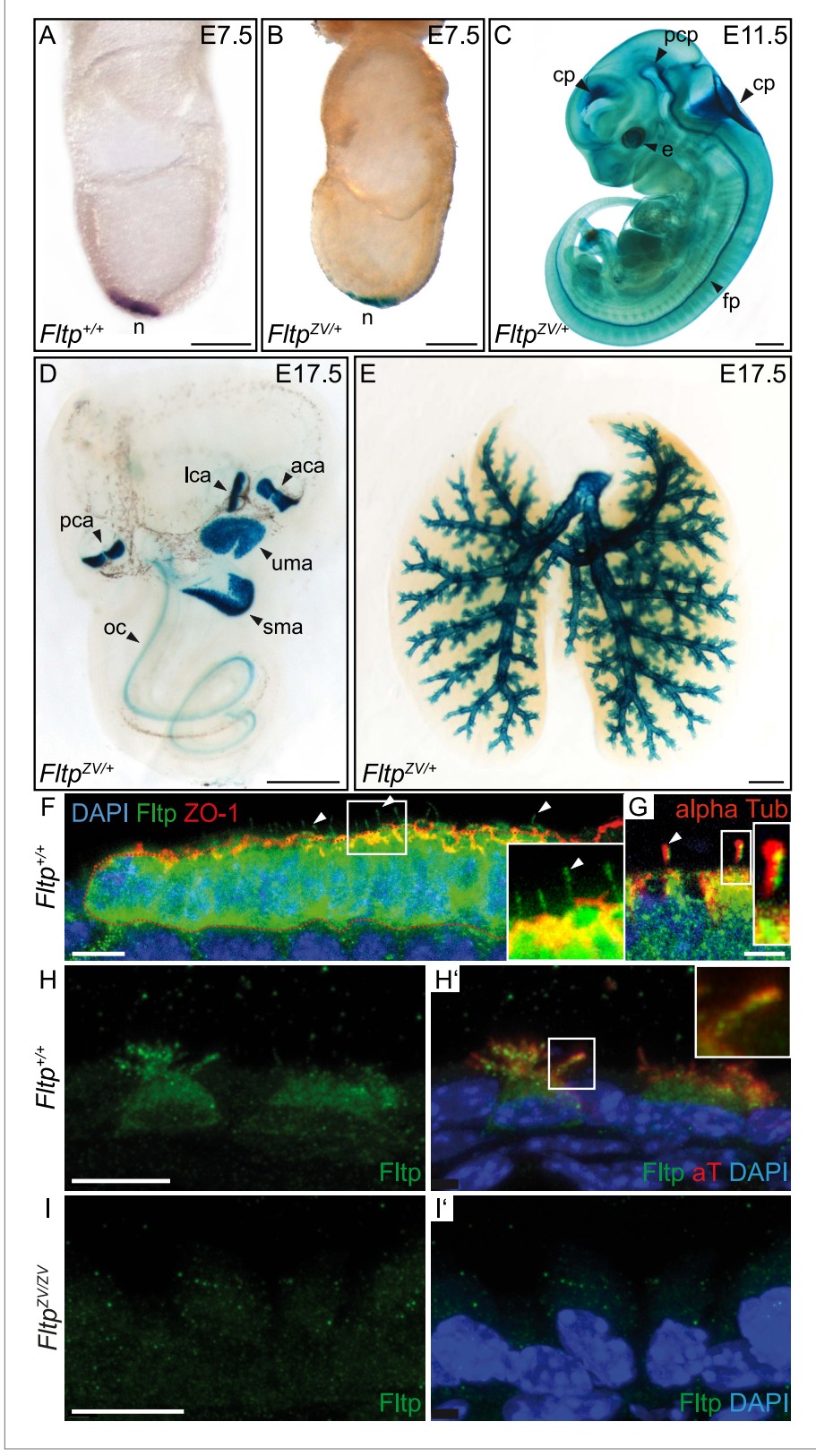

**Figure 2**. Fltp reporter and protein is detectable in mono- and multiciliated tissues. (**A** and **B**) mRNA (Fltp in situ hybridization) (**A**) and *lacZ* reporter expression (**B**) are restricted to the node (n) at E7.5. (**C–E**)Whole-mount lacZ stained and benzyl alcohol/benzyl benzoate (BABB) cleared *Fltp^{ZV/+}* embryo and organs. (**C**) E11.5 embryo reveals

*Figure 2. Continued on next page*

*Figure 2. Continued*

reporter expression in the choroid plexi (cp), prechordal plate (pcp), eye (e), and floor plate (fp). (**D**) E17.5 IE shows reporter expression in posterior crista ampullaris (pca), lateral crista ampullaris (lca), anterior crista ampullaris (aca), utricular macula (uma), and saccular macula (sma) of the vestibular part as well as in the organ of Corti (oc). (**E**) E17.5 lung shows lacZ reporter activity in multiciliated lung epithelial cells. (**F** and **G**) Whole-mount antibody stained embryo (node, E7.75) analyzed by LSM reveals Fltp protein in vesicular fashion in the cytoplasm and along primary cilia (white arrow heads). (**H–I'**) Immunohistochemistry on cryosections combined with LSM analysis reveals Fltp at the apical plasma membrane and at cilia in multiciliated lung epithelial cells of *Fltp$^{+/+}$* adult animals (**H** and **H'**). No Fltp immunoreactivity is detected in *Fltp$^{ZV/ZV}$* lungs (**I** and **I'**). ZO-1 marks apical TJs, α-Tubulin (alpha Tub), and acetylated-Tubulin (aT), the tubulin network and cilia; nuclei are marked by DAPI and Fltp by Fltp1 (**F** and **G**) and Fltp116-1 (**H–I'**). Scale bars; 100 μm (**A** and **B**), 500 μm (**C, D, E**), 10 μm (**F**), 4 μm (**G**), and 10 μm (**H–I'**).
The following figure supplement is available for figure 2:

**Figure supplement 1**. Additional *Fltp* reporter expression in mono- and multiciliated tissues.

expected Mendelian ratio and are viable and fertile on CD1, C57BL/6NCrl and 129S6/SvEvTac background (*Figure 1—source data 1*).

Next, we characterized the temporal and spatial expression pattern of the *Fltp* reporter during embryonic development. LacZ reporter activity in *Fltp$^{ZV/+}$* animals is first detectable in the embryonic node at E7.5 (*Figure 2B*), which accurately reflects endogenous mRNA expression (*Figure 2A*) and confirmed our previous findings (*Tamplin et al., 2008*; *Lange et al., 2012*). At E8.5–13.5, strong reporter activity is detected in the eye, notochord, floor plate of the neural tube, and all four choroid plexi (*Figure 2C*, *Figure 2—figure supplement 1A–D*), tissues that are either mono- or multiciliated and depend on active PCP signaling. Consistent with a potential function in the PCP pathway, *LacZ* reporter activity in the IE coincides with BB docking and kinocilium positioning at E13.5 (*Figure 2— figure supplement 1E*) (*McKenzie et al., 2004*; *Ezan and Montcouquiol, 2013*). The *Fltp* reporter remains expressed during development and early postnatal life in all six sensory regions of the IE (*Figure 2D*, *Figure 5A*, *Figure 2—figure supplement 1C,E,F*). Furthermore, *Fltp* reporter activity strongly correlates with the spatial and temporal onset of ciliogenesis in the lung trachea and the main bronchi at E14.5–17.5 (*Figure 2E*, *Figure 2—figure supplement 1G,H*). In adult lungs at postnatal day (P) 30, the dual *lacZ* and *H2B-Venus* reporter remains expressed in MCCs of the bronchioles, but not in non-ciliated cells of the alveoli (*Figure 2—figure supplement 1I,J*). To confirm the reporter studies, we analyzed cell type-specific and subcellular localization of the endogenous Fltp protein in the embryonic node and adult lung epithelium. Whole-mount antibody staining combined with laser scanning microscopy (LSM) on gastrula-stage embryos revealed that the Fltp protein is localized at the apical PM, TJs, primary cilia, and in the cytoplasm of node cells (*Figure 2F,G*). Immunohistochemistry on lung sections further shows that Fltp is highly enriched at the apical cortex and in cilia of the multiciliated bronchial epithelial cells (*Figure 2H,H'*). The absence of Fltp immunoreactivity in *Fltp$^{ZV/ZV}$* lung sections further confirmed antibody specificity (*Figure 2I,I'*). Taken together, *Fltp* reporter and endogenous expression accurately correlate with onset of PCP establishment in the embryonic node, floor plate, choroid plexi, IE, and lung, which is indicative for a function in or downstream of the PCP pathway.

## Fltp regulates BB docking and cilia formation in multiciliated lung cells

The embryonic and adult lung depends on PCP signaling for branching morphogenesis, BB docking, and cilia formation (*Yates et al., 2010*; *Vladar et al., 2012*). First, we investigated the highly stereotypic branching pattern in cleared lacZ stained lungs at P60 (*Metzger et al., 2008*). No difference was detectable in *Fltp$^{T2AiCre/+}$*; *R26$^{R/+}$*(Gt[ROSA]26Sor) controls lungs with normal levels of Fltp protein (*Soriano, 1999*; *Lange et al., 2012*), *Fltp$^{ZV/+}$* heterozygous and *Fltp$^{ZV/ZV}$* homozygous mutant lungs, indicating that Fltp has no function during branching morphogenesis (*Figure 2E*, *Figure 3A–C*). In contrast, measurement and quantification of terminal lung bronchiole diameters revealed a dose-dependent and statistically significant constriction of the distal airways (*Figure 3A–D*), suggesting that normal Fltp levels are important for lung airway homeostasis. To better understand the etiology of the airway constrictions, we analyzed *Fltp* expression and function on cellular level in *Fltp$^{+/+}$*, *Fltp$^{ZV/+}$*, and *Fltp$^{ZV/ZV}$* adult lungs at P7 and P30 (*Figure 3E–L*). *Fltp-driven H2B-Venus* reporter activity was confined to the nuclei of multiciliated epithelial cells in *Fltp$^{ZV/+}$* and *Fltp$^{ZV/ZV}$* lungs at P7 and P30. LSM

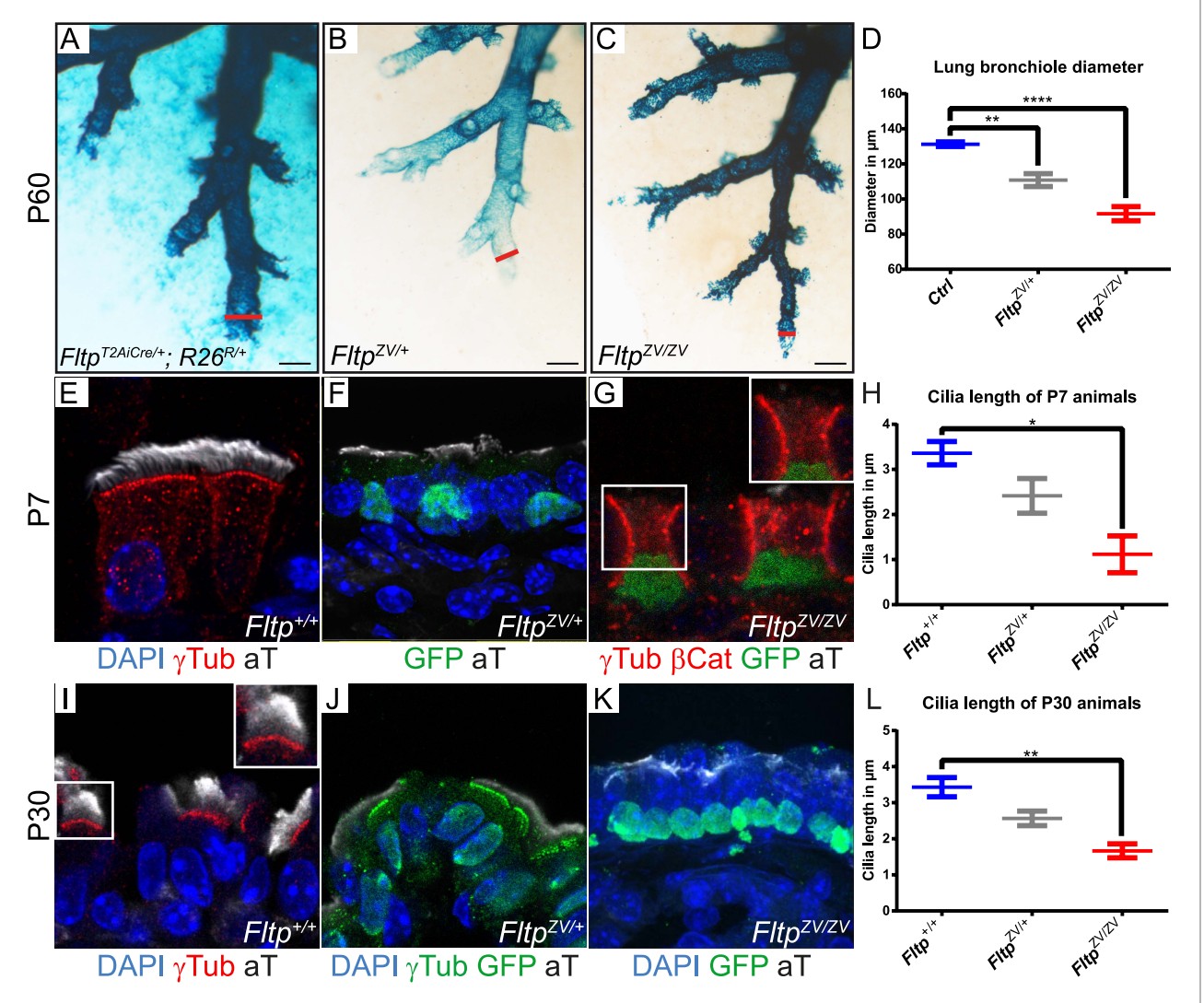

**Figure 3**. Loss of *Fltp* leads to constricted distal airways and cilia formation defects in the lung. (**A–C**) Whole-mount lacZ stained and BABB cleared distal airways of left lung lobes at P60. Red lines show how diameters were measured. (**A**) *Fltp^{T2AiCre/+}; R26^{R/+}*is used as *control* (*Ctrl*). (**B** and **C**) *Fltp^{ZV/+}* and *Fltp^{ZV/ZV}* animals show constricted distal airways. (**D**) Average distal airway diameter of *Ctrl* animals is 132.25 μm (n = 4; 125 bronchi), of *Fltp^{ZV/+}* animals is 99.68 μm (n = 3; 90), and of *Fltp^{ZV/ZV}* animals is 90.06 μm (n = 3; 95). (**E–G** and **I–K**) Immunohistochemistry on cryosections of lung distal airway epithelium combined with LSM analysis. (**E** and **I**) In *Fltp^{+/+}*animals BBs project cilia at the apical surface. (**F** and **J**) In *Fltp^{ZV/+}* animals less and shorter cilia are detectable. (**G** and **K**) *Fltp^{ZV/ZV}* animals often show absence of or shorter cilia at the apical surface. Note that only very few BBs are docked at the apical surface. (**H** and **L**) *Fltp^{ZV/ZV}* animals show significant shorter cilia than *Fltp^{+/+}* animals. In total we examined n = 3 (81 cells) for *Fltp^{+/+}*, n = 2 (22) for *Fltp^{ZV/+}*, and n = 2 (93) for *Fltp^{ZV/ZV}* P7 animals. For P30 we examined n = 3 (121 cells) for *Fltp^{+/+}*, n = 3 (134) for *Fltp^{ZV/+}*, and n = 2 (138) for *Fltp^{ZV/ZV}*. BBs are marked by γ-Tubulin (γTub), cilia by acetylated-Tubulin (aT), cell membrane by β-Catenin (βCat), nuclei by DAPI, and nuclei of *Fltp* reporter expressing cells by GFP. Statistical analysis uses an one way ANOVA (**p = 0.0028; ***p < 0.0001 for (**D**) and *p = 0.0143; **p = 0.0025 for (**H** and **L**)). Error bars show the 95% confidence interval of the mean (**D**) and the standard error of the mean (**H** and **L**). Scale bars; 200 μm (**A–C**).

analysis further confirmed that cilia formation was significantly affected in a dose-dependent fashion (***Figure 3E–L***), similar to PCP effector knock-down phenotypes in the mucociliary epithelium of *Xenopus laevis* (***Park et al., 2006***, ***2008***).

To analyze the dynamic cellular and molecular requirement for Fltp during BB docking, cilia formation, and PCP establishment, we employed the mTEC culture system (***You et al., 2002***; ***Vladar et al., 2012***). Switching mTEC culture to ALI conditions induces a differentiation process of lung progenitor cells towards a MCC phenotype. During differentiation the progenitor cells pass through different maturation stages (***Figure 4A***): Monociliated lung progenitor cells (stage I) start centrosome/BB

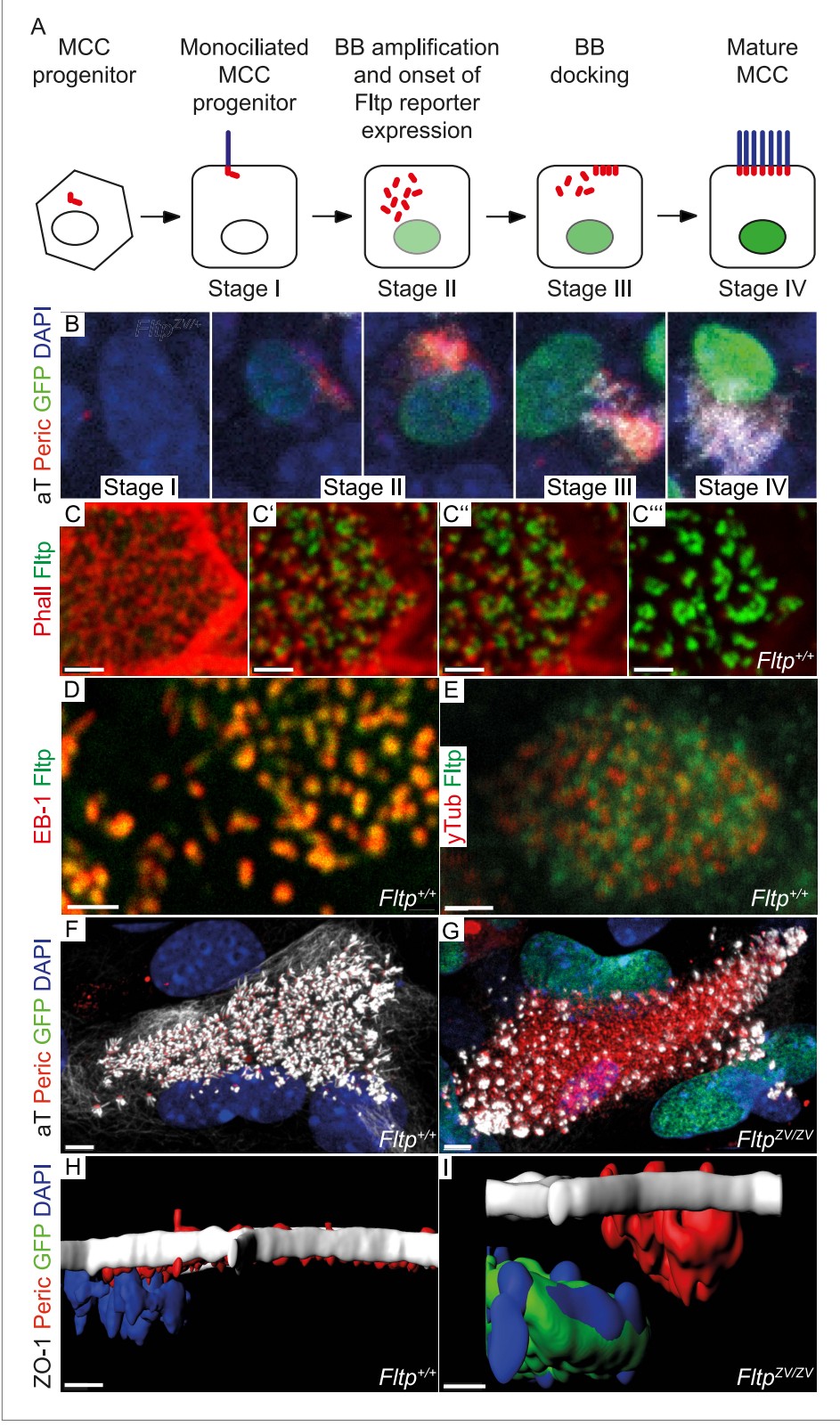

**Figure 4**. Fltp is expressed and necessary during BB docking. (**A**) Scheme of multiciliated cell (MCC) maturation. A MCC progenitor projects a primary cilium at the apical surface (stage I). Centrosome amplification is the first sign of differentiation (stage II) followed by apical transport and docking of BBs (stage III). Fully differentiated cells

*Figure 4. Continued on next page*

*Figure 4. Continued*

project multiple motile cilia at the apical surface (stage IV). Staging according to **Vladar and Stearns (2007)**. The green nucleus indicates *Fltp* reporter gene expression. (**B**–**G**) LSM of ALI culture of *Fltp^ZV* and *WT* mTECs. (**B**) Onset and level of *Fltp* reporter expression correlate with the onset of BB amplification and ciliogenesis (stage II–IV) in cultured ALI mTECs as shown in the scheme in (**A**). (**C**–**C'''**) Confocal sections of a single cell from the sub-apical actin level (**C**) over the apical actin level (**C'** and **C''**) to the cilia level (**C'''**) (ALI day 4). Fltp co-localizes with the MT-plus end binding protein EB-1 (**D**) (ALI day 4) next to the pericentriolar matrix (**E**) (ALI day 4) and to cilia in multiciliated *Fltp^+/+* cells (**C'''**). (**F**) *Fltp^+/+* cell showing all BBs (red dots) projecting cilia (white stripes) at the apical surface (ALI day 4). (**G**) In many *Fltp^ZV/ZV* cells the majority of BBs are not docked at the level of tight junctions marked by ZO-1 and do not project cilia (ALI day 4). (**H** and **I**) Side view IMARIS surface rendering shows that all BBs are docked at the apical surface in *Fltp^+/+* (**H**) (ALI day 4) in contrast to *Fltp^ZV/ZV* cells where most BBs stay in the cytoplasm (**I**) (ALI day 4). BBs are marked by γ-Tubulin (γTub) and pericentrin (Peric), cilia and the tubulin network by tyrosinated-Tubulin (Tyr Tub) and acetylated-Tubulin (aT), the actin network by Phalloidin (Phall), MT plus ends by EB-1, Fltp protein by Fltp116-1, nuclei by DAPI, and nuclei of *Fltp* reporter expressing cells by GFP. Scale bars; 2 μm (**C**–**C'''**, **D**, **E**), 5 μm (**F** and **G**).

The following figure supplement is available for figure 4:

**Figure supplement 1**. ALI differentiation efficiency and quantification of BB docking defect.

---

amplification followed by BB transport and docking (stage II–III) and subsequent cilia formation (stage IV) (**Vladar et al., 2012**). In these cultures, Fltp-driven H2B-Venus reporter activity is activated at the transition from stage I to stage II when centrosomes/BBs are amplified and docked in heterozygous and homozygous Fltp ALI cultures (**Figure 4B**). *Fltp-H2B-Venus* expression increases during maturation until cells are terminally differentiated. This correlates with active PCP signaling and initial asymmetric core component localization (**Vladar et al., 2012**). Subcellular co-localization studies further revealed that Fltp is localized to the apical, but not sub-apical actin cytoskeleton and fills the gaps in the actin network from where MT-based cilia project (**Figure 4C–C''',E**). Moreover, Fltp co-localizes with newly synthesized MT plus ends that are labeled with anti-EB1 antibodies (**Figure 4D**). Together with the direct adjacent localization of Fltp next to the pericentriolar matrix stained by γ-Tubulin (**Figure 4E**), these data suggest that Fltp connects BBs and ciliary MT plus ends to the cortical actin cytoskeleton. To test this idea, we examined mTECs switched to ALI conditions from *Fltp^+/+* and *Fltp^ZV/ZV* mice at day two and four of differentiation qualitatively and quantitatively. *Fltp^+/+* and *Fltp^ZV/ZV* differentiated with comparable efficiencies and speed as measured by Fltp-driven *H2B-GFP* expression and centrosome amplification (**Figure 4—figure supplement 1A**), and a comparable number of centrosomes are formed in differentiating *Fltp^+/+* and *Fltp^ZV/ZV* mTECs (**Figure 4F,G**), whereas BB docking and cilia formation was delayed at ALI+2 and +4 days of differentiation (**Figure 4F–I**, **Figure 4—figure supplement 1B**). Taken together, these data suggest that Fltp functions in the process of BB docking and cilia formation.

## Fltp regulates kinocilium positioning and stereocilia bundle morphogenesis in the inner ear

To analyze the function of Fltp in PCP, BB positioning, and cilia formation in vivo, we employed the best-characterized mammalian PCP model system, the organ of Corti in the IE. Morphologically the IE looked normal and the cochlear duct is not significantly shortened or widened in *Fltp^ZV/+* and *Fltp^ZV/ZV* mice, indicating that PCP-mediated convergent extension movements are not affected at P0 (data not shown). However, significant deviations of the polarized arrangement of stereocilia bundles can be seen in IHC and OHC rows with increasing severity closer to the apex along the apico-basal axis (**Figure 5B–C,J–N**). In addition, scanning electron microscopy (SEM) analysis uncovered severe stereocilia bundle morphogenesis defects and mispositioning of the MT-based kinocilium, very similar to phenotypes observed in mutants of spindle positioning proteins, such as Gα_{i3}, mPins, mInsc, and LGN (**Figure 5D–I**) (**Ezan et al., 2013**; **Tarchini et al., 2013**).

To seek first evidence if Fltp functions in the PCP pathway, we investigated a potential genetic interaction of *Fltp* with the core PCP component *Celsr1* (**Curtin et al., 2003**). As *Fltp* and *Celsr1* are co-expressed in the cochlea of the IE (**Shima et al., 2002**), we compared stereocilia bundle morphogenesis in *Celsr1^crsh/crsh* single mutant and *Fltp^ZV/ZV*; *Celsr1^crsh/crsh* double mutant cochleae at E18.5. On

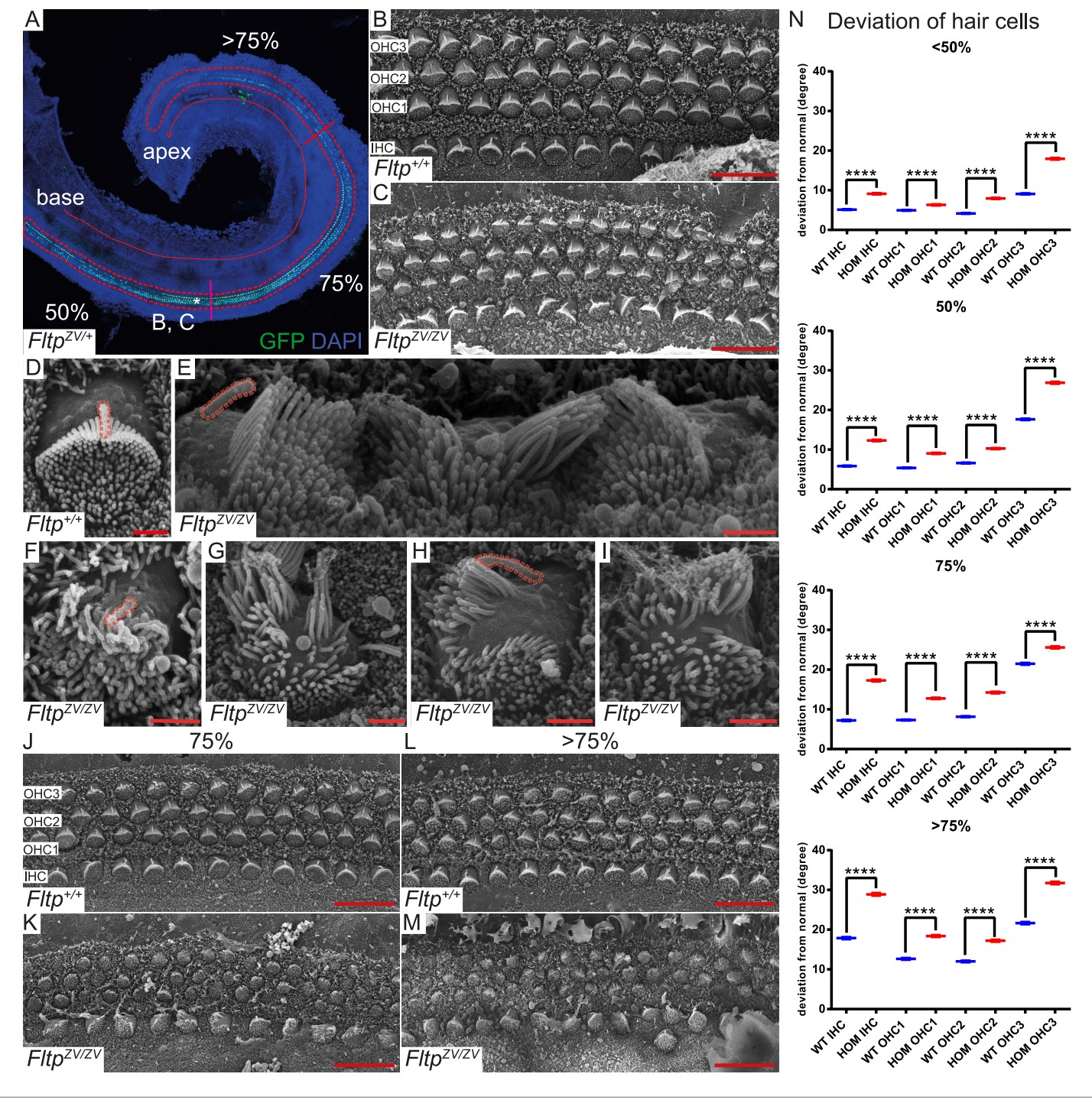

**Figure 5**. *Fltp* is required for kinocilium positioning and stereocilia bundle formation. (**A**) Whole-mount LSM of an *Fltp*$^{ZV/+}$ organ of Corti (OC) showing its different regions. *Fltp* reporter expression is restricted to all sensory HCs (IHC and OHC) of the OC (red dashed line). (**B–M**) Whole-mount SEM pictures of an E18.5 OC. (**B** and **C**) *Fltp*$^{+/+}$ (*WT*) HCs at 50% of the OC length are perfectly polarized (**B**) whereas *Fltp*$^{ZV/ZV}$ (*HOM*) HCs show orientational defects (**C**). (**D–I**) Enlargements of HCs reveal perfect polarized kinocilia as well as highly ordered stereocilia in *WT* (**D**). *HOM* cells are misaligned and show split stereocilia bundles (**E**, **G–I**), general bundle morphology defects (**E–I**), as well as detached kinocilia (**E**). Polarity and stereocilia bundles disruption increases in *HOM* hair cells from 75% (**K**) to >75% (**M**) of OC length compared to *WT* cells (**J** and **L**). (**N**) Deviation of HCs from the polarity axis for *WT* and *HOM* animals significantly differs in all OC regions (<50%, 50%, 75%, >75%). Quantification of HC rotation was performed by measuring the angle from the normal tissue polarity (measured by the medial to distal alignment of the HC rows) to the middle of the stereocilia bundle. In total we examined

*Figure 5. Continued on next page*

*Figure 5. Continued*

n = 4 (219 HCs) of *WT* and n = 7 (426) of *HOM* animals. Statistical analysis uses circular statistics (****p < 0.0001). Error bars show the standard deviation. Kinocilia are marked by the red dashed lines, nuclei by DAPI, and nuclei of *Fltp* reporter expressing cells by GFP. Abbreviations: inner hair cell (IHC), outer hair cell (OHC1-3). Scale bars; 10 μm (**B** and **C**), 1 μm (**D**–**I**).

the mixed genetic background analyzed, *Celsr1*$^{crsh/crsh}$ animals showed rather normal bundle alignment in IHCs and OHCs from the base to the middle part of the cochlea (<50%), but the alignment was significantly disturbed closer to the apex (>50%) (*Figure 6A,C,D*). In contrast, *Fltp*$^{ZV/ZV}$; *Celsr1*$^{crsh/crsh}$ double mutant cochleae revealed an increased degree of misaligned stereocilia bundles, most pronounced in the <50% region, and showed an additional fourth OHC row (*Figure 6B–D*). These data suggest that *Fltp* and the core PCP gene *Celsr1* either act together in the PCP pathway or in two parallel pathways important for the morphogenesis of the cochlea. To clarify if Fltp is necessary to establish PCP in the IE according to the current model (*Figure 6E*) (*Ezan and Montcouquiol, 2013*), we further investigated the localization of core PCP components. No differences in the asymmetric distribution of Vangl1 and Dvl2 at the cell cortex along the medio-lateral axis were detected in *Fltp*$^{+/+}$ and *Fltp*$^{ZV/ZV}$ cochleae at E18.5 (*Figure 6F–I*), consistent with a function of Fltp downstream of core PCP molecules.

## Dlg3 and Fltp regulate BB positioning in the inner ear

BB positioning and stereocilia bundle morphogenesis depend on spindle positioning proteins (*Ezan et al., 2013*; *Tarchini et al., 2013*), but how these connect to core PCP molecules and actin-rich stereocilia bundles is unknown. Due to PCP defects in the IE (*Van Campenhout et al., 2011*), we hypothesized that the mammalian scaffold protein Dlg3 might be a cortical adaptor for PCP-mediated BB positioning. Due to the lack of good antibodies and tools, we generated a transgenic mouse line that constitutively expresses a Venus fluorescent protein fused to the N-terminus of Dlg3 (Dlg3-Venus) (see Supplemental 'Materials and methods'). Next, we examined the subcellular localization of the Dlg3-Venus fusion protein in relation to BB, kinocilium, stereocilia bundle, core PCP molecules, and TJs in cochlea HCs using LSM and three-dimensional reconstruction (*Figure 7A–B',F–H*). Dlg3 specifically localizes in a crescent shape to the bare zone around the BB and co-localizes with the core PCP component Dvl2 at lateral TJs (*Figure 7A–B',G*), the area where mPins/LGN, mInsc, and Gα$_i$ are localized (*Ezan et al., 2013*; *Tarchini et al., 2013*). This suggests that Dlg3 might have a conserved function in spindle/BB positioning (*Johnston et al., 2009*). Interestingly, Fltp localizes at the interface between Dlg3 and the actin-based stereocilia bundles (*Figure 7C–E'*), further suggesting that Fltp might be an adaptor protein linking spindle/BB positioning proteins to the apical actin cortex.

We tested this idea by analyzing Dlg3 and BB localization in Fltp mutants. When compared to WT (*Figure 8A–D',J*, *Figure 8—figure supplement 1A*), the lateral crescent of Dlg3 in Fltp mutants is disturbed and BBs are not correctly positioned in the middle of the apical Dlg3 crescent (*Figure 8E–H',J*, *Figure 8—figure supplement 1A*), implicating that Fltp binds to Dlg3 to position BBs, kinocilia, and stereocilia bundles. If Dlg3 and Fltp cooperate to position BBs, one would predict that these proteins physically interact. Dlg3 contains a central SH3 domain which could bind to the SH3 binding or PRR domain of Fltp. We tested this by co-immunoprecipitation experiments of Streptavidin-Flag (SF)-tagged Dlg3 variants and Fltp-myc variants from serum starved and ciliated HEK293T cells (*Figure 9A,B*). These experiments suggest a physical interaction of Fltp with Dlg3, which requires the highly conserved N-terminal SH3 binding domain of Fltp, but not the SH3 domain of Dlg3 (*Figure 9B*). As Dvl2 localizes directly adjacent to Dlg3 and Fltp in the inner ear, we next examined the association of endogenous Dvl2 and γ-Tubulin with overexpressed SF full-length Fltp and SF-Dlg3 (*Figure 9C*). The interaction of Dvl2 with Dlg3 and Fltp and γ-Tubulin with Dvl2 and Dlg3 suggests that these proteins form a complex during BB transport and positioning. Finally, we tested the predicted requirement of Dlg3 for BB positioning by knock-out analysis (*Figure 8K,L*, *Figure 8—figure supplement 1B,C*). Taken together, these results suggest that Dlg3 is a BB positioning protein that is anchored at apical TJs (*Van Campenhout et al., 2011*) and cooperates with Fltp to position BB at the apical cortex of HCs in the cochlea at E18.5.

## Discussion

Activation of the PCP pathway results in rearrangement of the actin and MT cytoskeleton via tissue specific effector molecules. Although actin and MT interactions need to be thoroughly controlled in

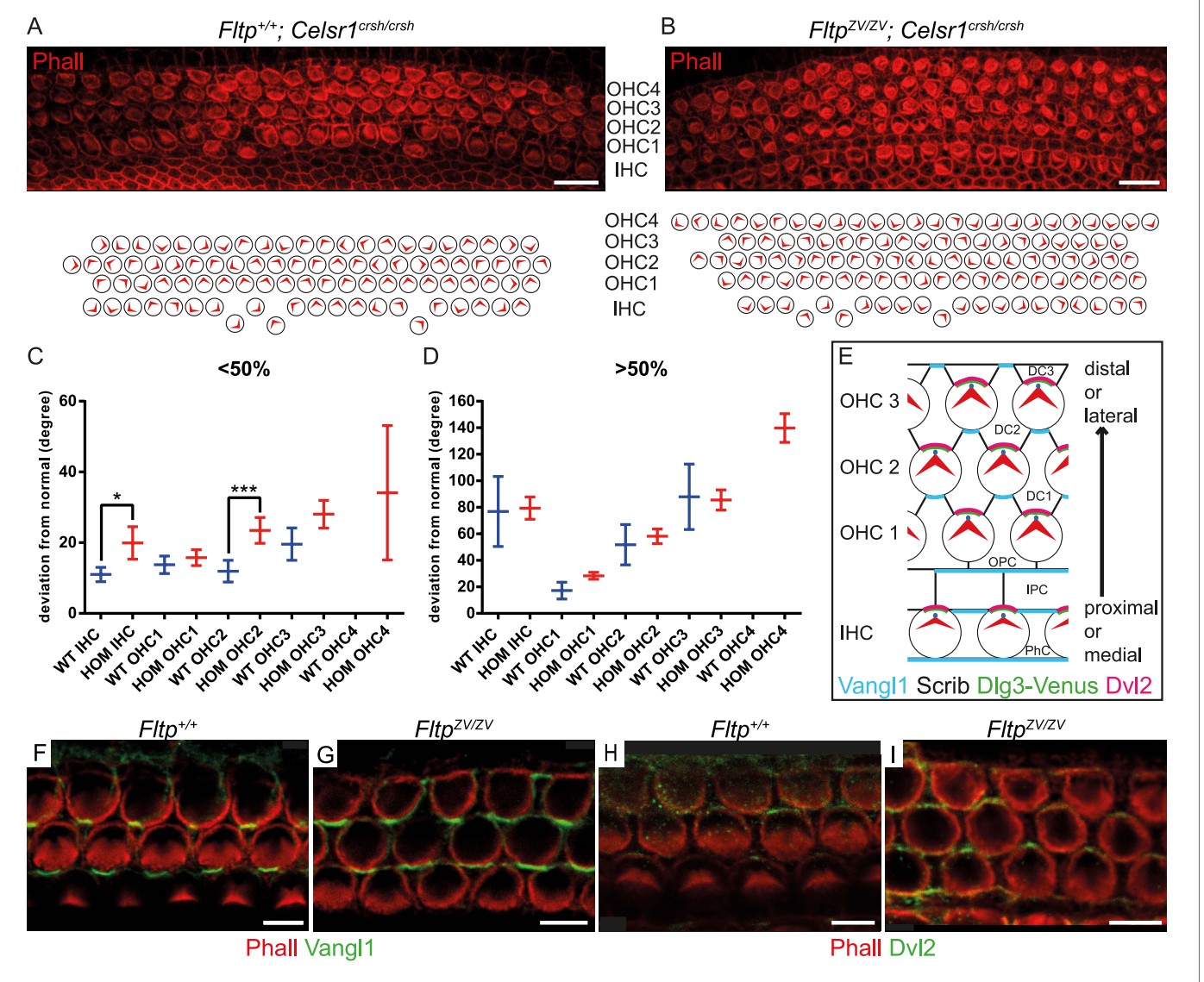

**Figure 6**. *Fltp* is a potential downstream mediator of PCP signaling. (A–B) LSM of an E18.5 *Fltp⁺/⁺; Celsr1^crsh/crsh* (WT) organ of Corti (OC) (**A**) revealed rotated outer (OHC) and inner hair cells (IHC). *Fltp^ZV/ZV; Celsr1^crsh/crsh* (HOM) OC (**B**) shows more severely rotated IHCs and OHCs as well as an additional OHC row in comparison to (**A**). (**C** and **D**) Hair cells of HOM (red) animals show a more pronounced PCP phenotype compared to WT (blue) in the region <50% and >50% of the OC (for cochlea region nomenclature and quantification method see *Figure 5*). In total we analyzed n = 1 (130 cells) at <50%, n = 1 (90) at >50% for WT, n = 5 (298) at <50%, n = 5 (743) at >50% for HOM. Statistical analysis uses a Kruskal–Wallis test (*p = 0.0375; ***p = 0.0003). Error bars show the 95% confidence interval of the mean. (**E**) Model illustrating PCP molecule localization in IE hair cells. (**F–I**) Whole-mount IE (E18.5) LSM of 3 OHC rows revealed Vangl1 localization at the lateral side of supporting cells in *Fltp⁺/⁺* (**F**) and *Fltp^ZV/ZV* (**G**) animals and Dvl2 localization at the lateral side of IE hair cells in *Fltp⁺/⁺* (**H**) and *Fltp^ZV/ZV* (**I**) animals indicating that core PCP protein localization is not disrupted. The actin network and stereocilia are marked by Phalloidin (Phall) and core PCP proteins by Vangl1 and Dvl2. Abbreviations: inner phalangeal cells (PhC), inner pillar cells (IPC), outer pillar cells (OPC), Deiters' cells (DC1-3). Scale bars; 10 μm (**A** and **B**), 5 μm (**F–I**).

virtually all cell types in different cell cycle phases across species from yeast to humans, it is surprising that not many adaptor proteins that mediate such interactions are identified (***Rodriguez et al., 2003***). These global cytoskeletal rearrangements underlie many fundamental morphogenetic processes in which cellular asymmetries need to be established and maintained, such as tissue morphogenesis, cell migration, or BB and spindle positioning in post-mitotic and mitotic cells, respectively (***Wallingford, 2012***). We describe the identification and functional analysis of Fltp, a protein that orchestrates cytoskeletal rearrangements in cells that acquire PCP. Our data suggest, that

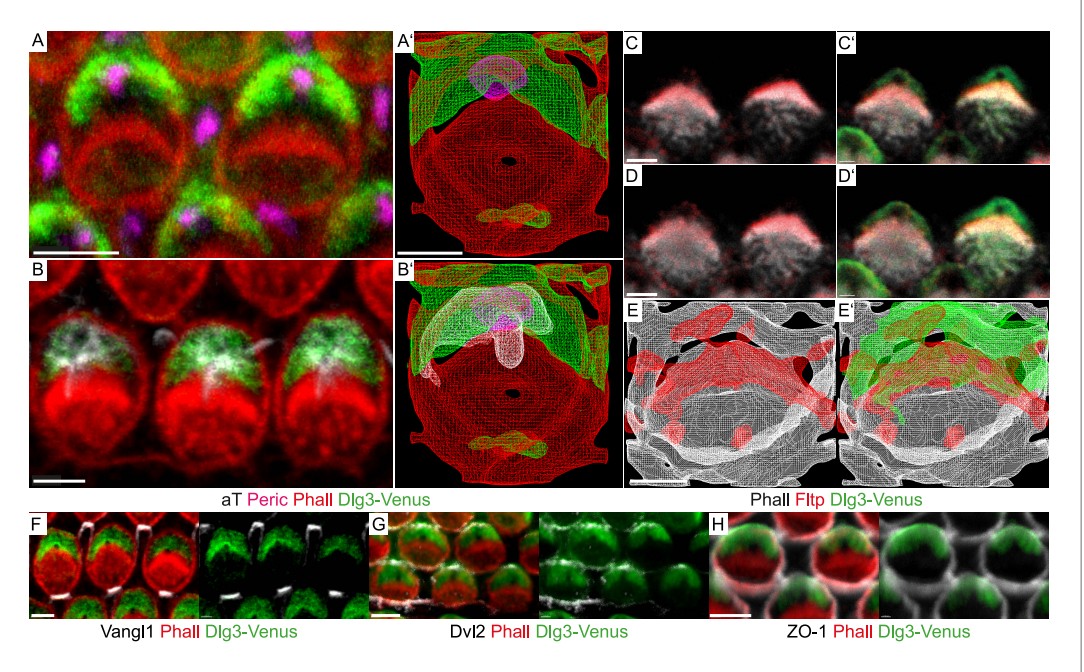

**Figure 7**. Fltp is located at the interface of apical actin and Dlg3 in IE hair cells. (**A**, **B**, **C–D′**, **F–H**) Single section LSM of outer HCs of an *Fltp⁺/⁺; Dlg3-Venus* animal at E18.5 reveals that Dlg3-Venus is located at the lateral membrane and at the medial membrane (or the lateral membrane of the supporting cell) of IE HCs (**A** and **B**). Fltp is localized lateral to the cuticular plate (CP) and the stereocilia bundles (SC) (**C** and **D**). Dlg3-Venus is located in a lateral crescent overlapping with Fltp localization (**C′** and **D′**). The unstained area marks the region of the BB (**C′** and **D′**). Dlg3-Venus is co-localized with Dvl2 (**G**) at the most lateral membrane directly opposite of Vangl1 (**F**) and with ZO-1 at the apical membrane (**H**). (**A′**, **B′**, **E**, **E′**) IMARIS wireframe animation of a *Dlg3-Venus* IE HC showing Dlg3-Venus co-localization with the BB, the kinocilium, and actin (**A′** and **B′**) and a *Fltp⁺/⁺; Dlg3-Venus* IE HC showing Fltp, Dlg3-Venus, and Phalloidin co-localization (**E** and **E′**). The actin network, the CP, and the SC are marked by Phalloidin (Phall), Fltp protein by Fltp116-1 (Fltp), the kinocilium by acetylated-Tubulin (aT), the BB by pericentrin (Peric), the apical cell membrane by ZO-1, core PCP proteins by Vangl1 and Dvl2, and Dlg3-Venus fusion protein by GFP. Scale bars; 2 μm (**A–E′**), 3 μm (**F–H**).

Fltp is important for BB docking and positioning in mammals through regulating actin and MT interactions.

## *Fltp* is a potential PCP effector gene that regulates MT–actin interactions in mammals

Several lines of evidence suggest that Fltp is a potential PCP effector regulating the cytoskeleton. First, the *Fltp* gene and reporter are expressed in several tissues known to depend on active PCP signalling, such as the node, ependymal cells, floor plate, lung, eye, and IE. Second, *Fltp* expression kinetic correlates with dynamic PCP acquisition, terminal differentiation, and cellular morphogenesis in the lung and the IE. Third, the double knock-out of *Fltp* and the core PCP gene *Celsr1* show a more severe PCP phenotype than both single mutants, raising the possibility that both genes act in the same pathway, which needs further confirmation. If this is the case, then Fltp seems to be a downstream effector molecule in the PCP pathway, as it is not required for PCP establishment, but rather for cell intrinsic PCP transduction leading to cytoskeletal rearrangements. This is supported by the fact that phenotypes observed for the PCP effectors Intu and Fuz in the mucociliary epithelium of the frog and Fltp in the murine lung are strikingly similar (*Park et al., 2006*, *2008*). Additionally, spindle positioning rather than core PCP mutant phenotypes mirrors the Fltp mutant phenotype in the IE (*Ezan et al., 2013*; *Tarchini et al., 2013*). Finally, Fltp is localized at the interface of spindle positioning complexes that capture MT plus ends and connect to the PM and the apical actin cytoskeleton. Loss of Fltp leads to failure in BB docking and positioning most probably through disconnection

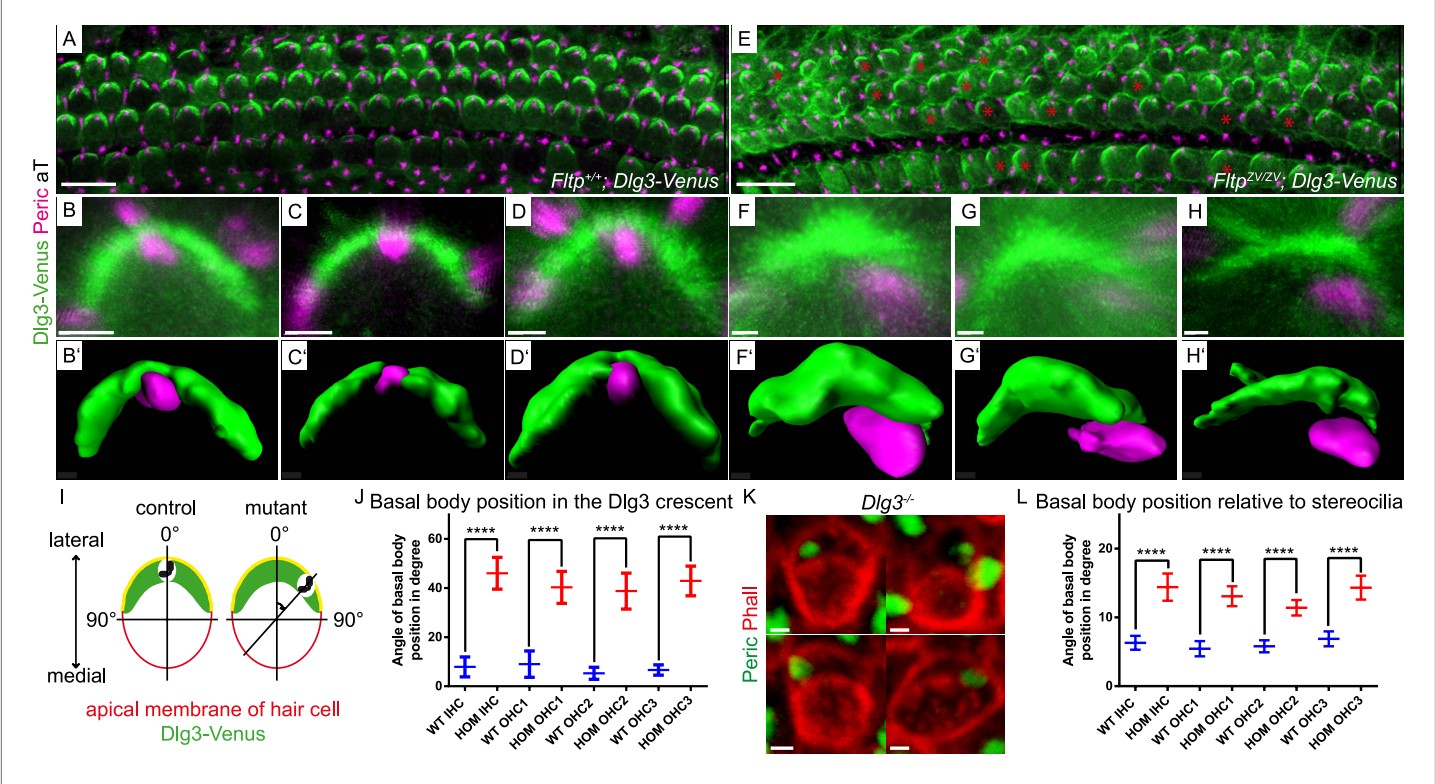

**Figure 8**. Loss of *Fltp* or *Dlg3* leads to BB mispositioning in IE hair cells. (**A**–**D**) LSM of *Fltp⁺/⁺; Dlg3-Venus* (*WT*) IE HCs at E18.5 reveals BBs in the middle of the lateral Dlg3-Venus crescent. (**B'**–**D'**) IMARIS surface rendering pictures of *WT* HCs. (**E**–**H**) LSM of *Fltp^ZV/ZV; Dlg3-Venus* (*HOM*) IE HCs at E18.5. BBs are located at the edge of the lateral Dlg3-Venus crescent. The crescent itself often shows defective localization. The red asterisk marks some affected cells. (**F'**–**H'**) IMARIS surface rendering pictures of *HOM* HCs. For quantification see *Figure 8—figure supplement 1A*. (**I**) For BB mispositioning analyses the angle between the middle of the Dlg3-Venus crescent and the BB location was measured. (**J**) BB position in affected HCs of *HOM* animals significantly differs from the position in the *WT*. In total we analyzed n = 2 (73 cells) for *WT* and n = 2 (119) for *HOM*. (**K**) LSM of *Dlg3⁻/⁻* (Dlg3^tm1Grnt) IE HCs reveals mislocalized BBs and rotated hair cells. For quantification *Figure 8—figure supplement 1B,C*. (**L**) Analysis of BB mispositioning was performed as described in (**I**). BB position in affected HCs of *Dlg3⁻/⁻* (*HOM*) animals significantly differs from the *Dlg3⁺/⁺* (*WT*) position. In total we analyzed n = 7 (470 cells) for *WT* and n = 13 (804) for *HOM*. Statistical analysis uses an one-way ANOVA (**J**) or a Kruskal–Wallis test (**L**) (****p < 0.0001). Error bars show the 95% confidence interval of the mean. The actin network and stereocilia are marked by Phalloidin (Phall), BBs by pericentrin (Peric), cilia by acetylated-Tubulin (aT), and Dlg3-Venus fusion protein by GFP. Abbreviation: inner hair cell (IHC), outer hair cell (OHC 1-3). Scale bars; 10 μm (**A** and **E**), 1 μm (**B**–**D'**, **F**–**H'**, **K**).

The following figure supplement is available for figure 8:

**Figure supplement 1**. BB mispositioning in *Fltp^ZV; Dlg3-Venus* and *Dlg3* animals.

of the BB, MT, and actin cytoskeleton in the mammalian lung and IE. Although the mechanisms of Fltp action in these two tissues might differ, a common theme seems that Fltp acts at the interface of the MT and actin cytoskeleton for BB docking and positioning, which we discuss in the following sections.

## Fltp regulates BB docking and cilia formation in the lung

PCP is best studied in the mammalian IE, but is also apparent in MCCs of the lung and mucociliary epithelium of the frog skin (*Wallingford, 2012*). In both terminally differentiated cell types, core PCP molecules are asymmetrically localized at the apical PM and translate long-range PCP signals into cell intrinsic cytoskeletal changes that coordinate BB docking and positioning as well as cilia formation in mono- and multiciliated cells. In MCCs of the frog skin the PCP effector proteins Intu and Fuz have been shown to control ciliogenesis and together with Dvl control apical docking and planar polarization of BBs by coordinating apical actin assembly (*Park et al., 2006*, *2008*). It was further suggested that the PCP effector Fuz and Dvl2 control membrane trafficking and vesicle fusion at the BB essential

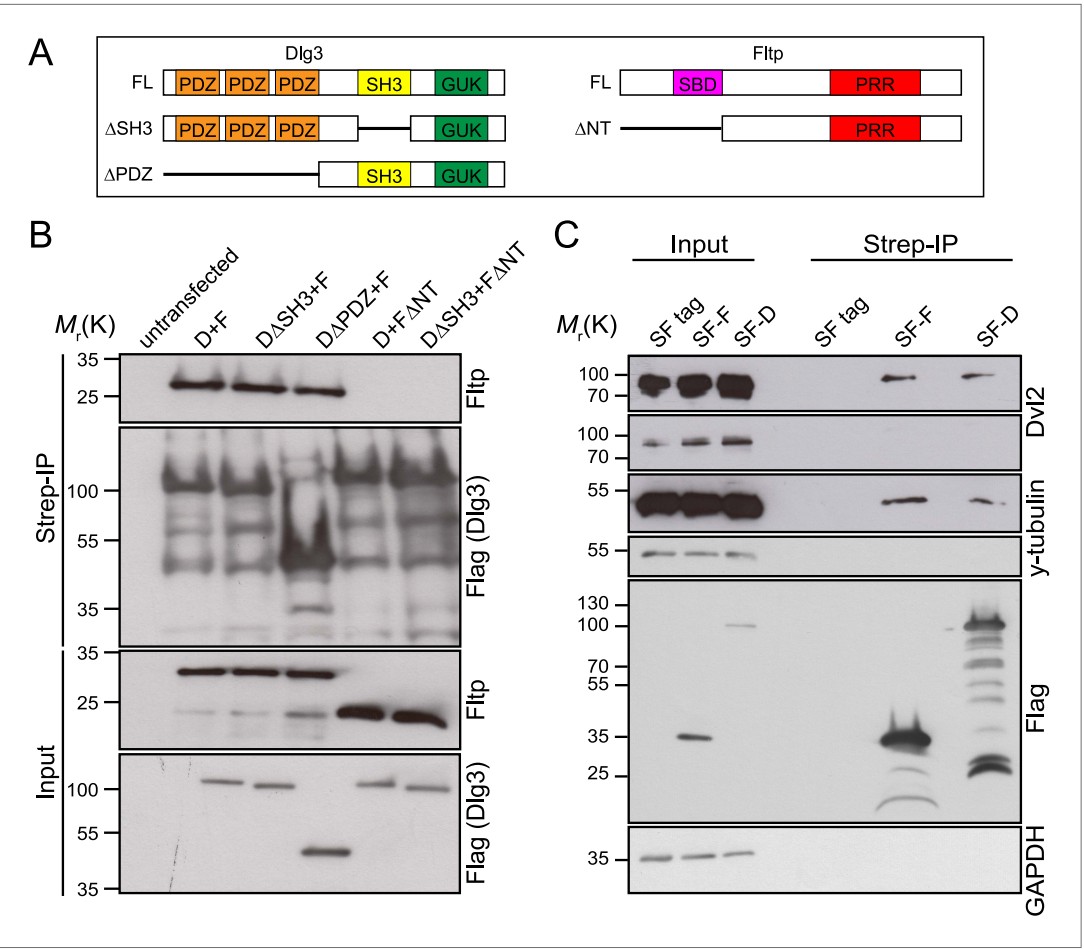

**Figure 9**. Fltp interacts with proteins associated with the TJ complex (Dlg3), the BB (γ-Tubulin) and with the core PCP protein Dvl2. (**A**) Dlg3 and Fltp constructs used for interaction domain mapping. (**B**) N-terminus of Fltp is essential for interaction with Dlg3. HEK293T cells were transfected with SF Dlg3 variants and with Fltp-myc variants. SF-tagged Dlg3 was immunoprecipitated using Streptavidin beads (Strep-IP). Full-length Fltp-myc was detected in the Strep-IP in the presence of full-length SF-Dlg3, SF-Dlg3ΔSH3, and SF-Dlg3ΔPDZ. FΔNT cannot be co-immunoprecipitated with SF-Dlg3. (**C**) Fltp and Dlg3 are found in a complex together with Dvl2 and γ-Tubulin. HEK293T cells were transfected with SF Dlg3 and SF Fltp. In a Strep-IP for Fltp and Dlg3, endogenous Dvl2 and γ-Tubulin were co-immunoprecipitated. Abbreviations: D: Dlg3; F: Fltp; SBD: SH3 binding domain; PRR: proline rich repeat.

for ciliogenesis (*Park et al., 2008*; *Gray et al., 2009*). Studying Fltp function in the ALI primary cell culture system allowed us to link the onset of *Fltp* expression to the time of BB amplification, docking, and cilia formation. In MCCs, we have shown a close association of Fltp with the apical actin–MT cytoskeleton and BBs. Fltp localization at the apical surface is reminiscent of Dvl, Sec8, and Intu localization in frog multiciliated mucociliary epithelium (*Park et al., 2006*). Lack of Fltp function leads to BB docking defects and reduced number of cilia formed at the apical PM in ALI cultures and in vivo. Fltp physically interacts with Dlg3 and the core PCP molecule Dvl2, which regulates BB transport together with proteins of the secretory pathway (*Park et al., 2008*). We have previously shown that Dlg3 is the only Dlg family member that is monoubiquitinated by Nedd4 and Nedd4-like E3 ligases. This is required for apical PM transport mediated by the motor protein Dynein IC and Sec8, a component of the exocyst complex (*Van Campenhout et al., 2011*). Thus, taken together, these findings suggest that Fltp cooperates with Dlg and Dvl family members to regulate apical BB transport and docking of BBs in multiciliated tissues. However, there is likely not a strong parallel between BB docking in MCCs and BB positioning in the IE, a topic which needs to be addressed in the future.

## Fltp and the spindle positioning protein Dlg3 act in concert to position BBs in the IE

To better understand the function of Fltp, we further focused our analysis on the IE as a well-described model for PCP. It was previously shown that core PCP molecules localize to distinct apical membrane compartments, but differential localization seems not sufficient to instruct morphogenesis of actin-rich hair bundles (*Jones and Chen, 2008*). Instead, BB as well as kinocilium formation and positioning is essential for hair bundle morphogenesis, which is regulated by opposing localization of spindle positioning proteins and apical polarity proteins (*Ezan et al., 2013*; *Tarchini et al., 2013*). mInsc, mPins/LGN, and Gα$_i$ localize in a microvilli-free zone (bare zone) at the lateral side, whereas the apical polarity complex consisting of Partitioning defective 3 and 6 (Par3, 6) as well as atypical protein kinase C (aPKC) localize at the medial side. Surprisingly, localization of spindle proteins was not affected in the classical PCP mutants (*Ezan et al., 2013*), which either suggests redundancy in the PCP pathway or alternative mechanisms that position spindle proteins and BBs in the IE. We provide evidence that Fltp and Dlg3 position BBs and stereocilia bundles in the IE in addition to Gα$_i$, mPins/LGN, and mInsc, suggesting that we have identified important new molecules for BB/spindle positioning in mammals. Dlg3 and Fltp together surround the BB and anchor this organelle asymmetrically to the apical cortex likely via Dvl2 and TJ-associated proteins on the one side and attachment to the apical actin cytoskeleton on the other side. We have recently shown that the scaffold protein Dlg3 is important for apical polarity and TJ formation by interaction with a number of TJ-associated proteins (*Van Campenhout et al., 2011*). In the fruit fly, Dlg, Mushroom body defective (Mud), and Pins are absolutely necessary for spindle positioning in various cell types in vitro and in vivo (*Bellaiche et al., 2001*; *Johnston et al., 2009*; *Bergstralh et al., 2013*). The co-localization of Gα$_i$, mPins/LGN and mInsc, and Dlg3-Venus in the bare zone further suggests that also in mammals these proteins act in concert to position BBs and spindles. Furthermore, the basolateral polarity complex consisting of Dlg, Lgl, and Scrib is known to establish membrane domains by reciprocal inhibitory interactions with the apical aPKC-Par3/6 polarity complex and to regulate A–B spindle positioning (*Knoblich, 2008*; *Siller and Doe, 2008*). These reciprocal inhibitory interactions of the apical and basolateral polarity complex at the apical surface further stabilized the positioning of BBs at the apical PM. But how spindle positioning complexes are linked to the actin cytoskeleton is still a mystery in mammals. In yeast, MT plus ends are captured by Kar9a, which binds directly to a type V myosin motor bound to actin filaments (*Korinek et al., 2000*; *Lee et al., 2000*). As described above, Fltp functions very likely as a molecular adaptor protein localized between the interface of spindle positioning complexes and the apical actin cytoskeleton. Loss of Fltp function leads to disconnection of MT-based kinocilium and actin-based stereocilia bundles. Together, this suggests that Fltp couples BB/spindle positioning proteins to core PCP molecules and to the actin cytoskeleton in the IE. Further studies have to be conducted to precisely clarify how this complex is anchored to the cytoskeleton and core PCP components.

## *Fltp* function in development and disease

Knock-out analysis of *Fltp* revealed an important function in the terminal differentiation of multiciliated lung cells and the morphogenesis of stereocilia bundles on IE HCs. This emphasizes that PCP acquisition and remodeling of the actin- and MT-based cytoskeleton is crucially important for the maturation and function of terminally differentiated cell-types in tissues and organs. Consequently, mutations of both core PCP and effector molecules lead to a wide spectrum of ciliary dysfunction syndromes (*Gerdes et al., 2009*; *Wallingford, 2012*). For instance, in primary cilia dyskinesia, lack of directed mucociliary clearance and recurrent respiratory tract infections can eventually progress to permanent lung damage (*Storm van's Gravesande and Omran, 2005*). The *Fltp* knock-out mouse shows BB docking and cilia formation defects in MCCs of the lung as well as terminal airway constrictions. This suggests that human *FLTP* might be mutated in lung disease, a hypothesis that we are currently testing with our clinical partners. Moreover, mutation of Fltp leads to a stereocilia morphogenesis defect in the cochlea of the IE, which suggests that mutation of *FLTP* can cause hearing loss in human. Mutations in human *DLG3* are associated with X-linked mental retardation (*Lickert and Van Campenhout, 2012*) and we have previously identified that the scaffolding protein Dlg3 regulates apical polarity and TJ formation as well as PCP in the IE (*Van Campenhout et al., 2011*). In this study, we have shown that Fltp and Dlg3 cooperate in BB positioning in the IE. As *Drosophila* Dlg acts as a tumor suppressor regulating proliferation and asymmetric cell division (*Johnston et al., 2009*), it is well possible that the Fltp–Dlg3 complex is involved in similar cellular processes in mammals. Indeed,

we have evidence that Fltp regulates cell division of intestinal stem cells (Böttcher and Lickert, in preparation), suggesting that we identified a novel molecule with brought implication for ciliary disease and stem cell-mediated tissue homeostasis.

## Materials and methods

### Animal data

Mouse keeping was done at the central facilities at HMGU in accordance with the German animal welfare legislation and acknowledged guidelines of the Society of Laboratory Animals (GV-SOLAS) and of the Federation of Laboratory Animal Science Associations (FELASA). Post-mortem examination of organs was not subject to regulatory authorization.

### Generation of Fltp antibodies

Fltp antibodies were generated as described previously (*Lange et al., 2012*). Two affinity purified polyclonal antibodies (Fltp1, Fltp116-1) against mouse Fltp using the peptide sequence: DNPD EPQSSHPSAGHT for Fltp1 and KPFDPDSQTKQKKSVTKTVQ for Fltp116-1 were raised in rabbit (Pineda, Berlin, Germany). The Fltp1 epitope locates to the less well conserved C-terminal PRR (*Figure 1C*, red empty box). The Fltp116-1 epitope (*Figure 1C*, red filled box) resides N-terminal to the Fltp1 epitope and is less conserved in human.

### Antibodies

Primary antibodies used were rabbit anti-Fltp1 (Pineda, Berlin), rabbit anti-Fltp116-1 (Pineda, Berlin), mouse anti-ZO-1 (33–9100: Invitrogen, Carlsbad, CA), mouse anti-α-Tubulin (T6199; Sigma), mouse anti-acetylated Tubulin (T7451; Sigma), mouse anti-γ-Tubulin (ab11316; Abcam), rabbit anti-β-Catenin (C2206; Sigma), chicken anti-GFP (GFP-1020; Aves Labs), rat anti-tyrosinated Tubulin (MAB1864; Millipore), rabbit anti-pericentrin (PRB-432C; Covance), rabbit anti-Vangl1 (HPA025235; Sigma), rabbit anti-Dvl2 (3216; Cell Signaling), and Alexa Fluor 546 Phalloidin (A22283; Invitrogen). Immunostainings were performed as described in the Supplemental 'Materials and methods'.

### Western blot

Western blot analysis was performed by standard procedures. Following antibodies were used; mouse anti-Flag (A8592; Sigma), rabbit anti-Fltp1 (Pineda, Berlin), rabbit anti-Fltp116-1 (Pineda, Berlin), rabbit anti-Dvl2 (3216; Cell Signaling), mouse anti-γ-Tubulin (ab11316; Abcam), mouse anti-GAPDH (CB1001; Merck Bioscience).

### X-gal (5-bromo-4-chloro-3-indolyl-β-D-galactoside) staining

β-gal staining of whole-mount embryos and organs were performed as previously described (*Liao et al., 2009*). Some tissues were further processed. Not BABB treated whole-mount embryos/organs were fixed, washed in PBS, and photographed. BABB treated embryos/organs were left in BABB for photographing.

### SEM analysis

For SEM, inner ears were fixed in 2.5% glutaraldehyde in cacodylate buffer and then treated using standard procedures.

### Generation of the Fltp^ZV targeting vector

The knock-in/knock-out construct was designed as shown in *Figure 3A*. 5' and 3' HR for the *Fltp* gene were amplified by PCR (449, 450, 451, 452) using a C57BL/6J BAC clone (RP23-333P11) as template. These two PCR products were subcloned into the pL254 vector (*Liao et al., 2009*). The resulting vector was digested with *HindIII*, *SpeI* and electroporated into electrocompetent EL350 bacteria containing the *Fltp* BAC clone to retrieve the *WT* sequence between PCR homology arms resulting in the *Fltp* retrieval vector. For cloning of the knock-in/knock-out cassette in pBKS⁻ 5' and 3' HR for the knock-in into the ATG of exon, two of *Fltp* were generated by PCR (453, 454, 455, 456) using the previously mentioned BAC as a template and subcloned into pBKS⁻ using the introduced restriction sites, resulting in pBKS⁻-Fltp-HomArms. The targeting vector was generated by ligating the *loxP* flanked neomycin (*neo*) resistance cassette (PL-452) (*Liu et al., 2003*) into the pBKS⁻-H2B-Venus-intron-SV40pA plasmid resulting in pBKS⁻-H2B-Venus-intron-SV40pA-loxP-bGHpA-neo-EM7-PGK-loxP (pBKS⁻-H2B-Venus-neo). The T2A sequence from *Thosea asigna* virus was introduced into the *NotI* site

of pBKS⁻-H2B-Venus-neo by annealing the following oligos 2A_fwd; 2A_rev, which created a *NotI* compatible overhang resulting in pBKS⁻-2A-H2B-Venus-neo. NLS-lacZ (nuclear localization signal-β-galactosidase fusion protein) was ligated into the pBKS⁻-2A-H2B-Venus-neo vector resulting in pBKS⁻-NLS-lacZ-2A-H2B-Venus-neo. To finish the minitargeting construct, we cloned pBKS⁻-NLS-lacZ-2A-H2B-Venus-neo into pBKS⁻-Fltp-HomArms (both cut with *NotI* and *SalI*). The minitargeting construct was cut out by *SacII* and *KpnI*, electroporated in EL350 bacteria, and introduced into PL254 via bacterial homologes recombination resulting in the final targeting construct (PL254-Fltp-NLS-lacZ-2A-H2B-Venus-intron-SV40pA-loxP-bGHpA-neo-EM7-PGK-loxP) which was confirmed by sequencing and is ready for electroporating into embryonic stem (ES) cells (after linearization by *AscI*).

## Generation of the Fltp knock-out mouse line and Southern blot

To generate targeted ES cells, we electroporated IDG3.2-F1 ES cells (C57BL/6J x 129S6/SvEvTac) (*Hitz et al., 2007*) with the *AscI*-linearized FltpZV targeting vector and neomycin resistant clones were selected using 300 µg/ml G418 (Invitrogen). Homologous recombination at the *Fltp* locus was confirmed by Southern blot analysis of *DraIII*-digested genomic DNA using the Fltp 5′-probe (620 bp) (5′ S Fltp FWD *XhoI*; 5′ S Fltp REV *XbaI*). 8 out of 100 clones (in total: 22 positive clones out of 289) showed homologous recombination. Due to restriction fragment length polymorphisms (RFLP) of the Bl/6J and 129S6 alleles and an isogenic Bl6 targeting vector, homologous recombination occurred preferentially on the Bl/6J allele and reduced the size of the restriction fragment from 16.443 bp to 11.469 bp. The 129S6 WT band is smaller in size than the Bl/6J WT band and is not targeted. Both ES cell clones gave birth to Δneo animals, but for the analysis we concentrated on one clone and founded the animal colony on this. After deletion of the neo cassette mice were backcrossed to C57Bl/6J to eliminate the *Cre* allele. Only those mice negative for the *Cre* allele were used for back-crossings to C57BL/6NCrl, 129S6/SvEvTac, or CD1 and further analyses were carried out with *Fltp*^ZVΔneo/+^ mice backcrossed several generations in C57BL/6NCrl, 129S6/SvEvTac, or CD1.

## Generation of the Dlg3:Venus mouse line

The Dlg3-Venus mouse line was generated by electroporating a pCAG-Venus-Dlg3 construct into IDG3.2-F1 ES cells (C57BL/6J x 129S6/SvEvTac). Positive ES cells were aggregated with CD1 morulae and the resulting chimeras gave germ-line transmission of the Dlg3-Venus allele.

## Isolation of mouse tracheal epithelial lung cells and ALI culture

Isolation of mouse tracheal epithelial cells (mTECs) and air liquid interface (ALI) culture was performed as described in Vladar and Brody (*Vladar and Brody, 2013*).

Mice were sacrificed and the trachea dissected out. The trachea was put into ice cold HamF12 medium supplemented with penicillin/streptomycin (P/S). Tissue surrounding the trachea was removed under the dissecting microscope in HamF12 medium. The trachea was digested in HamF12 medium supplemented with 1.5 mg/ml pronase (protease 14) at 4°C ON. On the next day the tube was put on ice, FBS (fetal bovine serum) was added to a concentration of 10% and the tube was inverted at least 12×. The tracheas were transferred into a new tube with HamF12 P/S 10% FBS medium and again inverted 12×. Tracheas of the same genotype can be pooled and centrifuged for 10 min at 4°C at 400×*g*. 200 µl HamF12 P/S per trachea were supplemented with 0.5 mg/ml DNaseI and 10 mg/ml BSA (bovine serum albumin), the trachea were resuspended and incubated for 5 min on ice. Subsequently, the tubes were centrifuged for 5 min at 4°C at 400×*g*. The cells were resuspended in MTEC basic medium (DMEM F12 [#21331-020; Invitro], 15 mM HEPES [1M stock], 3.6 mM sodium bicarbonate, 4 mM L-Glut [200 mM stock], P/S [100Glu× stock], Fungizone [500× stock]) with 10% FBS and incubated for 3–4 hr at 37°C. The supernatant was collected, centrifuged for 5 min at 400×*g*, and resuspended in 100–200 µl MTEC plus medium (MTEC basic, 10 µg/ml insulin (5 mg/ml stock), 5 µg/ml transferrin (5 mg/ml stock), 0.1 µg/ml cholera toxin (100 µg/ml stock), 25 ng/ml EGF (25 µg/ml stock), 30 µg/ml BPE, 5% FBS, 0.01 µM RA) for counting. 75.000 cells per cm² were seeded on a transwell filter in MTEC plus medium. The medium was changed every 2 days. When the cells were confluent the MTEC culture was changed to an ALI culture by removing the medium on top of the filter and providing only medium supply from the bottom.

## Immunoprecipitation

For the interaction analysis, we used HEK293T cells transfected with Fltp-TAP-TAG and as a control HEK293T cells only transfected with TAP-TAG.

The cells were rinsed with warm PBS and lysed with 1 ml ice cold lysis buffer (50 mM Tris/HCl, pH 7.4, 150 mM sodium chloride, 2 mM EDTA, pH 8, 1% Nonidet P-40, filtrate sterile) per 10 cm dish on ice. Cells were spinned down for 10 min at 10000×$g$ at 4°C. Lysates were cleaned by filtration through syringe filters (MILLEX GP, 0.22 µm, Millipore). 1 mg of the filtered lysate was incubated with 50 µl Strep-Tactin superflow resin (IBA) in 1 ml of lysis buffer for 1 hr at 4°C in an overhead tumbler. The protein–Strep-Tactin solution was spinned down for 30 s at 7000×$g$ and transferred to microspin columns (GE-Healthcare). The supernatant was spinned down for 5 s at 100×$g$, washed three times with 500 µl TBST (100 mM Tris/HCl, pH 7.4, 1.5 M sodium chloride, 1.0% Tween20), and centrifuged for 5 s, 100×$g$. The protein was eluted by 500 µl desthiobiotin elution buffer (IBA) and incubated for 10 min while mixing the resin for several times. The elute can now be used for western blot analysis.

## Immunohistochemistry on cryosections and whole-mount immunohistochemistry

Dissected tissues were fixed in 4% paraformaldehyde (PFA) and cryoprotected by incubation in a sucrose gradient for at least 1 hr each (5%, 15%, 30%). Tissues were frozen in OCT (optimal cutting temperature) after which immunohistochemical staining was carried out on 8- to 12-µmthick sections, mounted on glass slides. Briefly, sections were rehydrated in PBS, permeabilised for 10 min in 0.1 M glycine/0.1% Triton X-100 in PBS, and blocked for 1 hr in 5% donkey serum/PBS-Tween 0.1% (PBS-T). Finally, the sections were incubated with the primary antibody in blocking solution ON at 4°C. The slides were washed with PBS-T, incubated with the secondary antibody in PBS for 2 hr at RT, washed with PBS, incubated with DAPI, and finally mounted with ProLong Gold antifade reagent (P36930; Invitrogen). Whole-mount immunohistochemistry was performed as previously described (*Nakaya et al., 2005*). Briefly, embryos were isolated, fixed for 20 min in 2% PFA in PBS, and then permeabilized in 0.1% Triton X-100 in 0.1 M glycine pH 8.0. After blocking in 10% FCS, 3% goat serum, 0.1% BSA, 0.1% Tween 20 for 2 hr, embryos were incubated with the primary antibody ON at 4°C in blocking solution. After several washes in PBS-T, embryos were incubated with secondary antibodies in blocking solution for 3 hr. During the final washes with PBS-T, embryos were stained with DAPI, transferred into 40% glycerol, and embedded between two coverslips using 120 µm Secure-Seal spacers (S24737; Invitrogen) and ProLong Gold antifade reagent.

## Immunohistochemistry on inner ears

Whole-mount inner ears were isolated, fixed for 20 min in 4% PFA in PBS, then the cochlea was dissected out, permeabilized in 0.1% Triton X-100 in 0.1 M glycine pH 8.0. After blocking in 10% FCS, 3% goat serum, 0.1% BSA, 0.1% Tween 20 for 25 min, ears were incubated with the primary antibody ON at 4°C in blocking solution. After several washes in PBS-T, embryos were incubated with secondary antibodies in blocking solution for at least 3 hr. During the final washes with PBS-T, embryos were stained with DAPI. Alternatively ears were fixed/permeabilized for 10 min in 0.25% glutaraldehyde, 3.7% PFA, 3.7% sucrose, and 0.1% Triton X-100 in PHEM buffer (60 mM Pipes, 25 mM Hepes, 5 mM EGTA, 1 mM MgCl), the cochlea was dissected out and again fixed/permeabilized for 10 min in the same solution as above. After PBS washes blocking as described above was carried out. Finally, ears were embedded in a spacer between two cover slips.

## Immunohistochemistry on ALI cultures

Antibody staining of ALI cultures was performed by cutting out, washing (PBS), and fixing (ice cold MeOH, 4% PFA, or fixed/permeabilised in 0.25% glutaraldehyde, 3.7% PFA, 3.7% sucrose, and 0.1% Triton X-100 in PHEM buffer depending on the primary antibody used) the Transwell membrane. Subsequently, membranes were treated as described above. Finally, membranes were embedded in a spacer between two cover slips.

## Genotyping of mouse lines

Genotyping of the Fltp[ZV] mouse line (after Cre-mediated excision of the neomycin selection cassette) was performed using the forward primer 566 and 418 as well as the reverse primer 565. With an annealing temperature of 57°C and 35 cycles, this PCR amplified a WT product of 317 bp and a targeted of 387 bp.

For genotyping of the Dlg3:Venus mouse line, the forward primer 180 and the reverse primer 181 were used. A PCR performed with an annealing of 60°C and 32 cycles amplifies a product of 312 bp.

Genotyping of Dlg3 mouse line was performed using the forward primer 534 and the reverse primer 535 as well as the reverse primer 536. Amplification of 33 cycles with an annealing of 58°C yielded a WT product of 535 bp and a 215 bp product for the targeted allele. The gene trap clone P038A02 (R1 on a pure 129Sv6 genetic background) was obtained from the German Gene Trap Consortium. Dlg3tm1Grnt/Y male and Dlg3tm1Grnt/+ female mice on a C57Bl/6 background were genotyped as previously described (*Cuthbert et al., 2007*).

For genotyping of the Fltp$^{ZV}$; Celsr1$^{Crsh}$ mouse line, we first performed a PCR with the forward primer 779 and the reverse primer 780 resulting in a 321 bp band. Next, we purified the PCR product via the PCR purification kit and sequenced the product in both directions. The adenine of the WT sequence was replaced by a guanine in the mutated sequence. For the Fltp$^{ZV}$ genotyping, we used the protocol described above.

## Isolation of embryos and organs

Dissections of embryos and organs were carried out according to Nagy and Behringer ('*Manipulating the mouse embryo: a laboratory manual*'). Embryos were staged according to Downs and Davies (*Downs and Davies, 1993*).

## Tissue clearing with BABB

Tissues were dehydrated through a methanol/$H_2O$ series: 2 hr in 25% methanol/$H_2O$, 2 hr in 50% methanol/$H_2O$, 2 hr in 75% methanol/$H_2O$, and ON in 100% methanol. Finally the tissue was transferred to BABB for clearing.

## Whole-mount in situ hybridization

Whole-mount in situ hybridization was performed as previously described and Fltp mRNA was transcribed from a sequence-verified cDNA clone (*Tamplin et al., 2008*).

## Paraffin sections

After in situ hybridization or lacZ staining, the embryos were dehydrated via a methanol series (25%, 50%, 75%, 2 × 100%) for 10 min per step. To clear the embryos, we incubated 2× in xylol for 5–10 min (depending on the thickness of the specimen). For tissue penetration of the paraffin, the specimen was left in paraffin at 65°C ON. On the next day, they were transferred into fresh paraffin and incubated at 65°C ON. The embryos were orientated and embedded into a mold. The cooled down paraffin blocks were mounted onto a grid and sectioned on a microtome. The sections were mounted on glass slides, dried ON at 37°C, dewaxed in xylol (2 × 15 min), fixed with mounting medium, and covered with a cover slip. After one night at 4°C, the sections were ready for microscopical analyses.

## Histological staining of paraffin sections using Nuclear Fast Red (NFR)

First, paraffin sections on glass slides were dewaxed twice for 15 min in xylene. An alcohol row (100%, 90%, 80%, 70%, 1 min each) for rehydration followed and ended in $H_2O$. Afterward, the slides were dipped into NFR for 1 min and thoroughly washed with dist. $H_2O$. Then, a dehydration step followed within an afferent alcohol row (70–100%, see above). Finally, the slides were incubated twice in xylene for 15–30 min and once in Roti-Histol for 15–30 min. After incubation in Roti-Histol, slides were put on a paper towel, sprinkled with a few drops mounting medium, and covered with a cover slip.

## Statistical analysis

Statistical significance was calculated using Prism (GraphPad Software). Circular statistics were done using MATLAB and R.

## Image acquisition and analysis

Image acquisition was performed on a Leica DMI 6000 confocal microscope or on a Zeiss Lumar.V12 Stereo using an AxioCam MRc5 camera. Image analysis was performed with Leica LAS AF software, AxioVision (Zeiss), and Imaris 7.6.4 (Bitplane, Zürich, Switzerland).

## Oligonucleotides for cloning

5′ S Fltp FWD: 5′-NNNCTCGAGGGAGCCCTTACGCACACTTAAG-3′
5′ S Fltp REV: 5′-NNNTCTAGACGGGACATTAACTGCATCTTATCTGAGGTTG-3′
011: 5′-NNNACTAGTAGGTAAGTGTACCCAATTCGCCCTATAG-3′

012: 5'-NNNGGATCCACGCGTTAAGATACATTGATGAGTTTGGAC-3'
013: 5'-NNNTCTAGAATGGTGAGCAAGGGCGAGGAGCTGTTC-3'
014: 5'-NNNACTAGTTTACTTGTACAGCTCGTCCATGCCGAGAG-3'
025: 5'-NNNGCGGCCGCGCCACCATGCCAGAGCCAGCG-3'
026: 5'-NNNTCTAGACTTAGCGCTGGTGTACTTGGTGATGG-3'
340: 5'-NNNGCGGCCGCGCCACCATGAACCTTGAAGCTCGAAAAACAAAG-3'
341: 5'-NNNGGCGCGCCTTTTTGACACCAGACCAACTGGTAATGGTAGC-3'
449: 5'-NNNGGCGCGCCAGTCAGGAAGTGGAAGAGAAGAACACAG-3'
450: 5'-NNNAAGCTTACTAGTGTGGTGGAGTGCCTGTCTACATGTG-3'
451: 5'-NNNAAGCTTCACGACAGTCAAAGCTGCAATAGAAC-3'
452: 5'-NNNGGATCCGGTAATTTGGCAATTATAGAACTCAGGC-3'
453: 5'-NNNCCGCGGAGCAGACTTAACTATGTTGGGGAAACAGC-3'
454: 5'-NNNGTCGACGCGGCCGCTGTTTACACTTGTTGCCTGGCAACTG-3'
455: 5'-NNNGTCGACGGTCCTAGTCTAGCTGAGGTCCAGATC-3'
456: 5'-NNNGGTACCATGCTGTGGGAGTCACTGACATTCTTG-3'

## Oligonucleotides for genotyping

180: 5'-GTGAACCGCATCGAGCTGAAGG-3'
181: 5'-GAACTCCAGCAGGACCATGTG-3'
418: 5'-AGCCATACCACATTTGTAGAGG-3'
534: 5'-GGTCTCTGATGAAGCAGTGATTTTT-3'
535: 5'-TGATGACCCATAGACAGTAGGATCA-3'
536: 5'-CTAAAGCGCATGCTCCAGAC-3'
565: 5'-CAGCATGGCATAGATCTGGAC-3'
566: 5'-GAGGCTGACTGGGAACAATC-3'
779: 5'-ACAACCTTTGGGCTCTCG-3'
780: 5'-TATAGTCCCTCCGGACCTC-3'

## Acknowledgements

We are extremely grateful to A Theis and B Vogel for excellent technical assistance and mouse work, Neil Copeland for generously providing plasmids and bacterial strains for homologous recombination in bacteria, and Ralf Kühn for IDG3.2 ES cells. Special thanks goes to Christiane Fuchs (ICB Consulting) for the circular statistics. We are particularly grateful to Jennifer Murdoch for providing Celsr1crsh animals. This work was supported by an Emmy-Noether Fellowship and the European Union with the ERC starting grant CiliaryDisease dedicated to HL. For financial support, HL thanks the Helmholtz Society, German Research Foundation, and German Center for Diabetes Research (DZD e.V.). This work was funded (in part) by the Helmholtz Alliance ICEMED—Imaging and Curing Environmental Metabolic Diseases, through the Initiative and Networking Fund of the Helmholtz Association.

## Additional information

### Funding

| Funder | Grant reference number | Author |
| --- | --- | --- |
| Deutsche Forschungsgemeinschaft | Emmy-Noether Fellowship | Heiko Lickert |
| European Research Council | ERC CiliaryDisease | Heiko Lickert |

The funders had no role in study design, data collection and interpretation, or the decision to submit the work for publication.

### Author contributions

MG, AB, Conception and design, Acquisition of data, Analysis and interpretation of data, Drafting or revising the article; IB, Designed the Fltp knock out construct, Conception and design; SH, Helped to establish ALI culture, Contributed unpublished essential data or reagents; CVC, Drafting

or revising the article, Contributed unpublished essential data or reagents; MA, AW, Helped to analyse SEM data; SGNG, Provided the Dlg3 knock out animals; HL, Conception and design, Analysis and interpretation of data, Drafting or revising the article, Contributed unpublished essential data or reagents

## Ethics

Animal experimentation: Mouse keeping was done at the central facilities at HMGU in accordance with the German animal welfare legislation and acknowledged guidelines of the Society of Laboratory Animals (GV-SOLAS) and of the Federation of Laboratory Animal Science Associations (FELASA). Post-mortem examination of organs was not subject to regulatory authorization.

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
