## [Decision Letter]

Thank you for sending your work entitled “Flattop and Dlg3 cooperatively translate planar cell polarity cues for basal body positioning” for consideration at *eLife*. Your article has been favorably evaluated by Fiona Watt (Senior editor), Jeremy Nathans (Reviewing editor), and 3 reviewers.

The manuscript has been reviewed by three expert reviewers, and their assessments together with my own, forms the basis of this letter. I am including the three reviews (lightly edited) at the end of this letter, as there are many specific and useful suggestions in them that will not be repeated in the summary here.

All of the reviewers were impressed with the importance and novelty of your work. Overall, the experiments appear to be carefully executed. However, all three reviewers believe that the data is over-interpreted in many places, and that this detracts from the manuscript. In particular, the evidence for a direct connection with PCP is circumstantial. There are also significant questions related to the data presented in Figures 3 and 4, and Figure 4—figure supplement 1.

We would like to encourage you to resubmit a revised manuscript that addresses the specific issues raised in the reviews below. With respect to the writing, I can appreciate that in the current publishing environment one feels compelled to impress editors and reviewers, which leads to an almost irresistible temptation to over-state one's own conclusions. *eLife* is trying to restore some balance to the world of scientific writing; we welcome self-critical comments and we believe that such comments actually enhance the reader's ability to judge the science fairly.

I will offer two examples of scientific writing that I think illustrates the preceding point. They are from two papers on ubiquitin that form the core of the discoveries for which the authors shared the Nobel Prize.

Proposed role of ATP in protein breakdown: conjugation of protein with multiple chains of the polypeptide of ATP-dependent proteolysis.

Hershko A, Ciechanover A, Heller H, Haas AL, Rose IA. Proc Natl Acad Sci U S A. 1980 Apr;77(4):1783-6.

In the second paragraph of the Discussion section, after summarizing the evidence that leads the authors to propose the conjugation of APF-1 (=ubiquitin) to proteins as a way of marking them for degradation, the authors write : “Evidence that APF-1-proteins are intermediates in the breakdown of denatured protein as proposed in Figure 6 is indirect and inconclusive at this time.”

Activation of the heat-stable polypeptide of the ATP-dependent proteolytic system.

Ciechanover A, Heller H, Katz-Etzion R, Hershko A. Proc Natl Acad Sci U S A. 1981 Feb;78(2):761-5.

In the last paragraph of the Discussion section, after describing their discovery and characterization of ubiquitin ligase, the authors write “Evidence suggesting the role of the APF-1-activating enzyme in conjugation and in ATP-dependent protein breakdown is not conclusive at present.”

Reviewer #1:

In this work, Moritz Gegg et al. study Fltp mutant mice and present evidence that Fltp plays a role in cilia docking by interacting with Dlg3. Since the work is based on studies of mutant mice and cells, it is very strong and I like the study very much.

I regret that several results are a bit overstated and/or not fully supported by the morphological illustrations provided. The main point is to tone down some overstated claims at several places in the text. As it is written, the paper presents a fully demonstrated model that is supported by the data but not yet demonstrated beyond reasonable doubt. The last part of the Introduction is a good example of what I mean. The whole discussion is unfortunately hyperbolic along the same lines.

I understand that formal proof of the model cannot be provided using reagents available and therefore the authors should not be required to do additional experiments.

On the other hand, they should pay attention and significantly edit their text to make it less speculative and more factual. I also have some remarks about figures.

1) Introduction:

In MGI, five Dlg genes are mentioned, each has been inactivated separately and Dlg1 is clearly the most critical, with neonatal lethality; intriguingly, Dlg3 knock-outs are reported to be viable and fertile.

2) Results:

a) Authors mention the presence of two P-rich domains in Fltp. P-rich regions are not detected using “conserved domain” (Entrez website), and the concentration of P in the boxed in Figure 1 does not seem that high. What criteria are used to support the existence of those two domains?

b) For antibody specificity (Figure 1—figure supplement 1), is the alpha-tubulin a cross reaction? If yes, why are microtubules not seen in immunostained cells (Figure 1—figure supplement 1)? It should be explicitly mentioned that the Abs were validated by immunocytochemistry in transfected cells.

c) Figure 2—figure supplement 1' and associated text: I cannot see endogenous Fltp green localization in monocilia. Could the figures be improved?

d) Figure 4 and associated text: This requires better explanations and/or illustrations. The text and legend do not allow the reader to understand Figure 4. Similarly, it is not very clear that Fltp localizes along the length of cilia (B,B'). This being said, the defective docking of BB in culture is very convincing, more than what is described in vivo in Figure 3. How can this be explained?

e) Inner ear, Figure 5. From 5C, I have the impression that PCP in IHC is even more disturbed than in OHC, and seems to be confirmed in the quantification N. However, Fltp is said to be expressed on in OHC. Does this mean that Fltp has some non-cell autonomous effects, or that low expression is present in IHC?

f) For quantification, authors mention ANOVA; it is probably acceptable for relatively small angles, less than 40^o^ in this series of data. Still, authors should be aware that comparison of angles is usually done using circular statistics, because arithmetic mean of angles is not meaningful.

g) The addition of phenotypes in Celsr1+Fltp mutants is interesting and worth showing. On the other hand, I do not think that this indicates that both proteins act in the same PCP pathway; they could just as well act in two parallel pathways. I think the text should be modified to take this into account. I also do not understand why authors looked at the localization of Vangl1 rather than Vangl2; which has a stronger action in PCP - for which good Abs are available commercially.

3) Last part of Results and Figure 7.

The subtitle and the following text are too strong. The data shown in Figure 7 are really informative, yet colocalisation does indicate but no prove interaction. The change in localization upon inactivation of Fltp, illustrated in Figure 8, provide a stronger argument for interaction than simple colocalisation, and the Co-IP in HEK cells is also a strong indication. As authors know and although their data are strong, interactions should be demonstrated using techniques such as IP in vivo, or FRET and others. I agree that this is undoable given the reagents available. However, given this intrinsic limitation, strong claims are not acceptable and should be significantly tuned down.

4) As mentioned above, the Discussion, although scholarly and informative, is somewhat hyperbolic and should be more nuanced. I would also like to have the opinion of the authors on why the effect on cilia docking is present only in some cells, whereas Fltp seems to be a unique gene without paralog.

Reviewer #2:

The authors present data showing that Fltp has a function in localizing basal bodies in multiciliated airway cells and in cochlear hair cells. Fltp is expressed in many ciliated cells where the PCP pathway functions. In fltp mutant MCCs, BBs fail to dock, and in mutant cochlea, BB/kinocilia are mispositioned and stereocilia bundles are misoriented and distorted. Fltp localizes apically in both cell types. The authors also describe, using a mutant, a function for Dlg3 in orienting BB/kinocilia. Finally, they show that transfected Fltp and Dlg3 co-IP, and that gamma tubulin and Dvl2 are also pulled down. These data could perhaps make a reasonable story that Fltp is involved in BB positioning in several cell types. However, the manuscript is written in such a way to substantially over-interpret their data, and a number of important holes in the data exist. As an appropriately described piece of work, it would be an editorial decision as to whether the results merit publication in this journal.

The overreaching interpretation of the results is evident with regard to several points. The authors devote considerable discussion to how Fltp participates in linking tubulin and actin in these processes, based only on the localization data and the phenotype, without any ultrastructure or biochemistry to support these conclusions. This discussion should all be deleted from the main text, and relegated to speculation in the Discussion.

A considerable omission is the failure to examine a potential role for Dlg3 in MCCs. Is there any MCC phenotype in the mutant? If not, how should one interpret the docking defect in comparison to the hair cell phenotype? This highlights the problem that there is not a strong parallel between BB docking in MCCs and BB positioning in hair cells.

A third substantial and related problem is that the model is very unclear. The cartoon suggests that Fltp localizes to stereocilia, but the binding data suggest it should be found with gamma tubulin at the BBs. Finally, the authors discuss function at the TJs. Since MCCs don't have the equivalent of stereocilia, where is the commonality? In these cells, Fltp is seen in the apical meshwork, but the resolution is too low to make specific conclusions about exactly where. We know too little to extract any inference about commonality of Fltp function from the two cell types.

There is one very troubling piece of data that must be explained, and unsatisfactory resolution will invalidate the MCC findings. The IMARIS resonstructions in Figure 4 HI show mostly docked vs many undocked BBs, but it is normally easy to find cells in day 4 cultures with undocked BBs, as cells continue to differentiate for some time. Care must be taken that these are representative. Therefore the quantitative data are essential. The MTEC data from day2 and day4 shown in Figure 4—figure supplement 1 are at best, extremely confusing, and at worst, suggest the possibility that the results should be discarded. First, it is unclear what the authors are measuring. The number of BBs? The number or percent of docked BBs? Number of cells with docked BBs? This must be made clear. Furthermore, in this highly reproducible system, at ALI+2 days, there are only very few cells with multiple cilia, yet the authors' data seem to suggest that 80% of the cells are multiciliated (to completion??). In the best wildtype cultures, 80% of cells are never seen to become multiciliated. Furthermore, by day 4, the number has declined to 60%? Why? At this stage, the numbers should still be increasing. In light of this, how are we to interpret the mutant results? We need a description of what is being measured, as well as images to corroborate.

Another problem is with the “genetic interaction” between fltp and celsr1. Genetic interaction is typically shown between heterozygotes that have no phenotype on their own, or between one het with no phenotype and a homozygote with a weak phenotype. The authors make double homozygous mutants and show an enhanced phenotype. Yet each homozygote has a phenotype of its own, and therefore the result could simply be an additive phenotype. The result therefore DOES NOT show a genetic interaction and provides NO EVIDENCE that Fltp is in the PCP pathway. The result is equally consistent with distinct pathways converging on a shared output. Numerous other statements describing Fltp function in PCP signaling are also misleading. For example, describing Dvl2 function at BBs as a PCP function is not valid. Indeed, the MCC phenotype is not a PCP phenotype at all, and the hair cell phenotype impacts PCP, but is presumably quite downstream. Taken together, it is not reasonable to describe the results as being effectors of PCP. This is another big example of interpretation not correctly reflecting the findings.

Reviewer #3:

This paper reports the discovery of a novel gene that acts as an essential downstream effector of PCP function in ciliated cells. Overall the paper is quite interesting and if correct would provide a good addition to the literature in this area, though there are several issue that must be resolved before publication can be considered. Moreover, many conclusions are overstated, and unnecessarily so, since the story is a good one even without the overreach.

The paper identified flattop in a screen for targets of foxa2, and this new gene is expressed in ciliated cells. Mutant mice display defects in the airway cilia as well as defects in kinocilium polarity in the cochlea, and these are traced to defects in basal body positioning. In the cochlea, fltp shows genetic interaction with the core PCP protein celsr1. The authors then show that the tight junction protein Dlg3 localizes to an interesting pattern in the cochlear hair cells and that fltp and dlg3 functionally and physically interact to control basal body position.

1) Many references are made to “Detailed” subcellular co-localization studies. But these data are not at all clear in their current form. In particular the data in Figure 3 are over-interpreted. It is stated that Fltp “specifically” colocalizes with newly synthesized MT plus-ends, but Figure 3 shows much green where there is no red and much of the converse as well.

2) The claim is made in Figure 3 that fltp is localized to the apical but not the sub-apical actin in multiciliated cells, yet Figure 3 shows only a single x/y image and Figure 3 shows Zo1, but not actin. How is this claim justifiable from these data?

3) The authors need to comment on the possible role for fltp in basal body amplification. According to Figure 5, the most obvious phenotype in the mutants may not be BB docking, but the reduction in BB numbers. This issue is a big one because recent work on cyclin O from Kinter and Omran shows that BB docking and amplification may be linked.

---

## [Author Response]

Reviewer #1:

1) Introduction: *In MGI, five Dlg genes are mentioned, each has been inactivated separately and Dlg1 is clearly the most critical, with neonatal lethality; intriguingly, Dlg3 knock-outs are reported to be viable and fertile*.

The reviewer is right and we corrected this in the manuscript. However, Dlg3 KOs show low penetrance embryonic lethality which we have published recently (Van In mammals, four Dlgs et al., 2011; Dev Cell): “From E8.5 onward, Dlg3 mutants are statistically underrepresented, and mutants displayed incompletely penetrant defects in embryonic turning, failure of chorioallantoic fusion, posterior truncations (n = 16 out of 38), and lack of anterior neural induction (n = 6 out of 38; Figure 1). Dlg3tm1Grnt null embryos also show occasionally an open brain phenotype (n = 8 out of 38; Figure 1—figure supplement 1). Taken together, both the Dlg3GtP038A02 and Dlg3tm1Grnt mutant alleles cause embryonic lethality with low penetrance.”

Change:

“In mammals, four Dlgs have been identified and we have recently shown that their function has diverged during evolution (41).”

To:

“In mammals, five Dlgs have been identified and we have recently shown that their function has diverged during evolution (41).”

2) Results:

*a) Authors mention the presence of two P-rich domains in Fltp. P-rich regions are not detected using “conserved domain” (Entrez website), and the concentration of P in the boxed in*
Figure 1
*does not seem that high. What criteria are used to support the existence of those two domains?*

For identification of the PRRs we used the protein domain prediction tools (MyHits, HYPERLINK “http://hits.isb‐sib.ch/cgi‐bin/motif_scan” \o “http://hits.isb‐sib.ch/cgi‐bin/motif_scan”http://hits.isb‐sib.ch/cgi‐bin/motif_scan and Scansite, HYPERLINK “http://scansite.mit.edu” \o “http://scansite.mit.edu/”http://scansite.mit.edu). The C-terminal PRR (138-184) was identified by MyHits.Author response image 1.

Unfortunately confusion occurred about the N-terminal PRR. The domain is now annotated as a SH3 binding domain identified by Scansite. However, PRRs are also known to be SH3 binding domains. We apologise for the mistake and we updated this in the manuscript and the figure (see below). We found Grb2 as an interaction partner in a Fltp interactome study indicating that this predicted Grb2 SH3 binding domain might be functional. Additionally, we showed that the N-terminal SH3 binding region of Fltp is important to bind Dlg3 (a SH3 domain containing protein, see Figure 9).Author response image 2.

Changes:

“An alignment of the proteins of different species reveals a high conservation during evolution as well as N- and C-terminal proline-rich repeats (PRR) (Figure 1).”

To:

“An alignment of the proteins of different species reveals a high conservation during evolution as well as a N-terminal SH3 binding domain and a C-terminal proline-rich repeat (PRR) (Figure 1).”

“The mouse and human proteins are highly homolog (clear green boxes: predicted proline rich repeats (PRRs); red filled box: peptide sequence of the Fltp116-1 epitope; red empty box: peptide sequence of the Fltp1 epitope; dark blue indicates conservation over 80%; lighter colors indicate less conservation).”

To:

“The mouse and human proteins are highly homolog (yellow box: SH3 binding domain; green box: predicted proline rich repeat (PRR); red filled box: peptide sequence of the Fltp116-1 epitope; red empty box: peptide sequence of the Fltp1 epitope; dark blue indicates conservation over 80%; lighter colors indicate less conservation).”

Abbreviations: D: Dlg3; F: Fltp

To:

Abbreviations: D: Dlg3; F: Fltp; SBD: SH3 binding domain; PRR:Proline rich repeat.

“The Fltp1 epitope lies in the PRR of the less well conserved C-terminal part of the Fltp protein (Figure 1, empty box). The Fltp116-1 epitope (Figure 1, filled box) resides N-terminal to the Fltp1 epitope and is less conserved in human.”

To:

“The Fltp1 epitope locates to the less well conserved C-terminal PRR (Figure 1, red empty box). The Fltp116-1 epitope (Figure 1, red filled box) resides N-terminal to the Fltp1 epitope and is less conserved in human.”

*b) For antibody specificity (*Figure 1—figure supplement 1*), is the alpha-tubulin a cross reaction? If yes, why are microtubules not seen in immunostained cells (*Figure 1—figure supplement 1*)? It should be explicitly mentioned that the Abs were validated by immunocytochemistry in transfected cells*.

We apologise for the confusion regarding the alpha-Tubulin staining. The membrane was first probed with anti-Fltp antibodies, developed and subsequently with anti-alpha Tubulin antibodies to confirm equal loading. We state this in the new Figure legend (Figure 1—figure supplement 1) and added that the antibodies were validated by immunocytochemistry in transfected cells to the manuscript.

Changes:

“We confirmed the specificity of these antibodies in Western blot analysis and immunocytochemistry (Figure 1—figure supplement 1 and Figure 1—figure supplement 2).”

To:

“We confirmed the specificity of these antibodies in Western blot analysis and immunocytochemistry in transiently transfected HEK293T cells (Figure 1—figure supplement 1 and Figure 1—figure supplement 2).”

“Anti-alpha-Tubulin antibody was subsequently used on the same blot to confirm equal loading.”

*c)*
Figure 2—figure supplement 1
*and associated text: I cannot see endogenous Fltp green localization in monocilia. Could the figures be improved?*

We improved the quality and magnified endogenous Fltp cilia staining on monociliated node cells and multiciliated lung cells in Figure 2.

*d)*
Figure 4
*and associated text: This requires better explanations and/or illustrations. The text and legend do not allow the reader to understand*
Figure 4*. Similarly, it is not very clear that Fltp localizes along the length of cilia (B,B'). This being said, the defective docking of BB in culture is very convincing, more than what is described in vivo in*
Figure 3*. How can this be explained?*

We are sorry for the inadequate description of Figure 4. We have added an illustration (Figure 4) and a better explanation in the text.

Briefly, this figure describes how Fltp-H2B-Venus reporter expression is activated upon the differentiation of lung progenitor cells into multiciliated cells. Different stages (I-IV) can be used to categorise the chronologic events during ALI culture differentiation (44). There is a clear correlation between Fltp-H2B-Venus reporter expression and amplification of BBs, the first visible step of progenitor differentiation to MCCs (Figure 4). We changed the text in the following way:”Upon switch to ALI culture, lung progenitor cells (stage I) differentiate into multiciliated epithelial cells by passing through stages of centrosome/BB amplification, BB docking (stage II-III) and cilia formation (stage IV). Fltp-driven H2B-Venus reporter activity is activated during centrosomes/BB amplification and docking (Figure 4), correlating with active PCP signaling and the onset of asymmetric localization of core components (42).”

To:

“Switching mTEC culture to ALI conditions induces a differentiation process of lung progenitor cells towards a MCC phenotype. During differentiation the progenitor cells pass through different maturation stages (Figure 4): Monociliated lung progenitor cells (stage I) start centrosome/BB amplification followed by BB transport and docking (stage II-III) and subsequent cilia formation (stage IV) (44). In these cultures, Fltp-driven H2B-Venus reporter activity is activated at the transition from stage I to stage II when centrosomes/BBs are amplified and docked in heterozygous and homozygous Fltp ALI cultures (Figure 4). Fltp-H2B-Venus expression increases during maturation until cells are terminally differentiated. This correlates with active PCP signaling and initial asymmetric core component localization (42).”

“(A-G) LSM of ALI culture of FltpZV mTECs. (A-B’) Single cell from the apical actin level (A, B) to the cilia level (A’, B-B’) (ALI day 9). Fltp connects to the apical actin, apical tyrosinated-Tubulin and localizes along cilia in multiciliated Fltp+/+ cells (A-A’, B-B’). Fltp is localized at the apical TJ level (C) and co-localizes with BBs (D) (ALI day 4). (E) Onset and level of Fltp reporter expression correlates with onset of BB amplification and ciliogenesis in cultured ALI mTECs. Staging according to (44).”

To:

“(A) Scheme of multiciliated cell (MCC) maturation. A MCC progenitor projects a primary cilium at the apical surface (stage I). Centrosome amplification is the first sign of differentiation (stage II) followed by apical transport and docking of BBs (stage III). Fully differentiated cells project multiple motile cilia at the apical surface (stage IV). Staging according to (44). The green nucleus indicates Fltp reporter gene expression. (B-G) LSM of ALI culture of FltpZV and WT mTECs. (B) Onset and level of Fltp reporter expression correlates with onset of BB amplification and ciliogenesis (stage II-IV) in cultured ALI mTECs as shown in the scheme in (A). (C-C’’’) Confocal sections of a single cell from the sub-apical actin level (C) over the apical actin level (C’,C’’) to the cilia level (C’’’) (ALI day 4). Fltp co-localizes with the MT-plus end binding protein EB-1 (D) (ALI day 4) next to the pericentriolar matrix (E) (ALI day 4) and to cilia in multiciliated Fltp+/+ cells (C’’’).”

We agree that Fltp localisation at the cilium is not perfect. We removed B and B’

*e) Inner ear,*
Figure 5*. From 5C, I have the impression that PCP in IHC is even more disturbed than in OHC, and seems to be confirmed in the quantification N. However, Fltp is said to be expressed on in OHC. Does this mean that Fltp has some non-cell autonomous effects, or that low expression is present in IHC?*

In the legend of Figure 5 we wrote that Fltp reporter expression is restricted to sensory HCs. This includes IHCs as well as OHCs. We changed this in the manuscript to make it clearer. Currently we have no reason to believe that Fltp might act non-cell autonomously.

Change:“Fltp reporter expression is restricted to sensory HCs of the OC (red dashed line).”

To:

“Fltp reporter expression is restricted to all sensory HCs (IHC and OHC) of the OC (red dashed line).”

*f) For quantification, authors mention ANOVA; it is probably acceptable for relatively small angles, less than 40*^*o*^
*in this series of data. Still, authors should be aware that comparison of angles is usually done using circular statistics, because arithmetic mean of angles is not meaningful*.

We thank the reviewer for pointing this out. In fact, we were not aware that a proper analysis should be done using circular statistics. Most angles are less than 40 degrees, but there are some exceptions. We hence repeated the testing procedure, this time using the tools of circular statistics. Our first approach was to replace the standard ANOVA (for two groups) by a Watson-Williams test. This is readily implemented in MATLAB and R. However, both implementations require equal group sizes. This was not given in our data. We hence simply performed a two-sided t-test. In the test statistic, we calculated the mean and the standard deviation according to circular statistics. The resulting p-values turned out to be even lower than in our original analysis. In fact, they often were below detection limit. Our results hence change such that all differences between wild types and homozygotes are significant. We implemented the new statistics in Figure 5 N.

“Statistical analysis uses a one way ANOVA (*** P=0.0001; **** P<0.0001). Error bars show the 95% confidence interval of the mean.”

To:

“Statistical analysis uses circular statistics (**** P<0.0001). Error bars show the standard deviation.”

*g) The addition of phenotypes in Celsr1+Fltp mutants is interesting and worth showing. On the other hand, I do not think that this indicates that both proteins act in the same PCP pathway; they could just as well act in two parallel pathways. I think the text should be modified to take this into account*.

We agree with reviewer 1 and 2 that Celsr1 and Fltp could act in two parallel pathways. We changed this in the manuscript.

Changes:

“To directly test if Fltp functions in the PCP pathway, we investigated genetic interaction of Fltp with the core PCP component Celsr1 (5).”

To:

“To seek first evidence if Fltp functions in the PCP pathway,we investigated a potential genetic interaction of Fltp with the core PCP component Celsr1 (5).”

“This demonstrates that Fltp genetically interacts with the core PCP gene Celsr1 and acts in the PCP pathway.”

To:

“These data suggest that Fltp and the core PCP gene Celsr1 either act together in the PCP pathway or in two parallel pathways important for the morphogenesis of the cochlea.”

*I also do not understand why authors looked at the localization of Vangl1 rather than Vangl2; which has a stronger action in PCP - for which good Abs are available commercially*.

Thank you for the comment but we were not aware that good Vangl2 antibodies are available. To analyse PCP in lung MCCs and tracheal epithelium, [42] used Vangl1 antibodies and could demonstrate nice asymmetric localisation. Fukuda et al. (2014) additionally showed asymmetric Vangl1 distribution in the inner ear. Based on these findings we used Vangl1 as a marker to show asymmetric core PCP protein distribution.

3) Last part of Results and Figure 7.

*The subtitle and the following text are too strong. The data shown in*
Figure 7
*are really informative, yet colocalisation does indicate but no prove interaction. The change in localization upon inactivation of Fltp, illustrated in*
Figure 8*, provide a stronger argument for interaction than simple colocalisation, and the Co-IP in HEK cells is also a strong indication. As authors know and although their data are strong, interactions should be demonstrated using techniques such as IP in vivo, or FRET and others. I agree that this is undoable given the reagents available. However, given this intrinsic limitation, strong claims are not acceptable and should be significantly tuned down*.

We changed the subtitle and tuned down the statements accordingly.

4) As mentioned above, the Discussion, although scholarly and informative, is somewhat hyperbolic and should be more nuanced.

We partially rewrote the Discussion to tune down the overstated conclusions.

*I would also like to have the opinion of the authors on why the effect on cilia docking is present only in some cells, whereas Fltp seems to be a unique gene without paralog*.

This is a very interesting question for which we have no definitive answer. The most likely explanation is the presence of other genes with similar functions, such as the PCP effector molecules Intu and Fuz (see Discussion).

We have initially concentrated our analysis on the lung and inner ear. Closer inspection of other cell types might reveal additional phenotypes, e.g. multiciliated ependymal cells were not investigated. In very rare cases (<1%) we observed an exencephalus phenotype indicative of a PCP-related phenotype. In Figure 1–figure supplement 3B Mendelian ration shows a slight underrepresentation of Fltp 129 KO animals, which we note in the figure legend. It is highly speculative, but left-right defects due to inappropriate ciliary position at the node might cause congenital heart defects. We will investigate this possibility in future experiments.

Reviewer #2:

*The authors present data showing that Fltp has a function in localizing basal bodies in multiciliated airway cells and in cochlear hair cells. Fltp is expressed in many ciliated cells where the PCP pathway functions. In fltp mutant MCCs, BBs fail to dock, and in mutant cochlea, BB/kinocilia are mispositioned and stereocilia bundles are misoriented and distorted. Fltp localizes apically in both cell types. The authors also describe, using a mutant, a function for Dlg3 in orienting BB/kinocilia. Finally, they show that transfected Fltp and Dlg3 co-IP, and that gamma tubulin and Dvl2 are also pulled down. These data could perhaps make a reasonable story that Fltp is involved in BB positioning in several cell types. However, the manuscript is written in such a way to substantially over-interpret their data, and a number of important holes in the data exist. As an appropriately described piece of work, it would be an editorial decision as to whether the results merit publication in this journal*.

*The overreaching interpretation of the results is evident with regard to several points. The authors devote considerable discussion to how Fltp participates in linking tubulin and actin in these processes, based only on the localization data and the phenotype, without any ultrastructure or biochemistry to support these conclusions. This discussion should all be deleted from the main text, and relegated to speculation in the Discussion*.

We agree with the reviewer and toned down or deleted over-statements.

*A considerable omission is the failure to examine a potential role for Dlg3 in MCCs. Is there any MCC phenotype in the mutant? If not, how should one interpret the docking defect in comparison to the hair cell phenotype? This highlights the problem that there is not a strong parallel between BB docking in MCCs and BB positioning in hair cells*.

The aim of our study was not to compare BB docking in MCCs and BB/cilia positioning in inner ear hair cells, but rather to provide a description of the Fltp phenotype in the lung and inner ear. We focused more on cellular and molecular mechanism of Fltp function in the inner ear, as it is a well-described PCP model system (as stated in the new discussion). Due to the Fltp phenotype in BB positioning, we focused in this system on Dlg3, as Drosophila Dlg is a well-known spindle positioning protein.

For this reviewer we examined a potential role of Dlg3 in MCCs. We analysed lungs of Dlg3 KO animals and could not detect an obvious difference in BB docking or cilia formation between KOs and WT animals (n=2). This indicates that Dlg3 and Fltp cooperate in the inner ear to position BB/cilia and other Dlg family members might compensate for Dlg3 loss-of-function in the lung MCCs. We provide this information for the reviewer only, because this goes beyond the scope of our manuscript.Author response image 3.(A-B) Show cryosections of a WT lung stained with acetylated tubulin (red) and pericentrin (green) and DAPI (blue). BBs are docked and cilia formed. (C-D) Cryosections of a Dlg3 KO animal show docked BBs and cilia. Scale bars: (A,C) 7μm, (B,D) 2μm.

*A third substantial and related problem is that the model is very unclear. The cartoon suggests that Fltp localizes to stereocilia, but the binding data suggest it should be found with gamma tubulin at the BBs*.

We apologise and removed the model from the figure set.

Finally, the authors discuss function at the TJs.

We discuss Dlg3 function at the TJ in the context of the IE (Figure 7), because Dlg3 was shown to localise to apical TJs and interact with TJ-associated proteins ([41]; Dev Cell).

*Since MCCs don't have the equivalent of stereocilia, where is the commonality? In these cells, Fltp is seen in the apical meshwork, but the resolution is too low to make specific conclusions about exactly where. We know too little to extract any inference about commonality of Fltp function from the two cell types*.

We now provide higher quality pictures of the localization of Fltp in the apical meshwork (Figure 4). As stated above, we do not intend to extract commonality of Fltp function in mono- and multiciliated cell types. However, the obvious commonality is the localization and close association of Fltp with the apical actin and MT cytoskeleton, the BB(s) and cilia (compare Figures 4 and 7) and the function in BB docking and positioning (compare Figures 4, 5 and 8).

*There is one very troubling piece of data that must be explained, and unsatisfactory resolution will invalidate the MCC findings. The IMARIS resonstructions in*
Figure 4
*HI show mostly docked vs many undocked BBs, but it is normally easy to find cells in day 4 cultures with undocked BBs, as cells continue to differentiate for some time*.

We agree with the reviewers comment that in day 4 cultures cells with undocked BBs can be found. This is exactly what we observed, except that Fltp homozygous cultures always show a reduced number of docked BBs compared to WT cultures.

*Care must be taken that these are representative. Therefore the quantitative data are essential. The MTEC data from day 2 and day 4 shown in*
Figure 4—figure supplement 1
*are at best, extremely confusing, and at worst, suggest the possibility that the results should be discarded. First, it is unclear what the authors are measuring. The number of BBs? The number or percent of docked BBs? Number of cells with docked BBs? This must be made clear*.

We apologize for the bad representation of our data; we measured number of cells with docked BBs (amount of docked BBs was categorized in different percentage groups). As this is too complicated we simplified the quantification and took care that we analysed representatives.

ALI cultures do not always differentiate with same rate and efficiency. To normalize for variability in differentiation we only compared WT and HOM cultures when they differentiated to a comparable extend (Figure 4—figure supplement 1). The percentage of differentiating MCCs; according to GFP/pericentrin staining among all cells analysed is shown.

Next, we examined BB docking by staging the differentiating cells into stage II-III (BB amplification and docking) and stage IV (every BB has a cilia and is docked) (see model Figure 4). This revealed at day 2 and day 4 BB docking defects in Fltp knock-out cultures, when compared to WT (Figure 4—figure supplement 1).

Representative images and how we quantified these are shown below:Author response image 4.(A-A’’) WT cultures at day 0, ALI day 2 and ALI day 4. In day 0 cultures MCCs are already present (A). Day 2 and day 4 ALIs show mostly 100% docked BBs (red, pericentrin; white, cilia) (A’,A’’). (B-B’’) In HOM cultures at day 0 some cells are already multiciliated but not to the same extend as WT cultures. At day 2 most stage II-III cells do not have docked their BBs yet. Significantly more BBs are docked in WT cultures (A’-A’’) than in HOM cultures (B’-B’’). Light white staining marks the intracellular MT network. Only bright white staining marks cilia.Scale bars; (A,B) 5μm; (A’,A’’) 25μm; (B’,B’’) 30μm.

*Furthermore, in this highly reproducible system, at ALI+2 days, there are only very few cells with multiple cilia, yet the authors' data seem to suggest that 80% of the cells are multiciliated (to completion??). In the best wildtype cultures, 80% of cells are never seen to become multiciliated*.

We only analysed cells differentiating into MCCs and excluded non-differentiating cells present in the ALI culture. The overall percentage of cells that started differentiating according to anti-GFP and anti-Pericentrin staining can be seen in the new Figure 4—figure supplement 1 (max. 22%). We categorized these cells into cells in stage II-III (BB amplification and docking) and stage IV (every BB has a cilia and is docked) in Figure 4—figure supplement 1.

Additionally, we analysed mTECs at ALI day 0 (directly before switching) and day 1. We could show that MCCs are already present before switching to ALI conditions. We propose that cells which are already multiciliated at day 0 develop to fully mature stage IV MCCs by day 2 accounting for this high amount of stage IV cells at day 2 (Figure 4—figure supplement 1).

*Furthermore, by day 4, the number has declined to 60%? Why? At this stage, the numbers should still be increasing. In light of this, how are we to interpret the mutant results? We need a description of what is being measured, as well as images to corroborate*.

This is a very good question for which we do not have a definitive answer. MCCs are already present at day 0 and 1. From day 0 to day 2 the amount of stage II-IV cells increases from approx. 5% to 18%. We hypothesize that this rapid increase in stage II-IV cells is due to the presence of MCCs before the switch to ALI conditions. These cells might also more efficiently show BB docking and cilia formation. Additionally, new stage I cells are entering the differentiation process after switch to ALI conditions. The amount of these cells increases and potentially reduce the number of docked stage IV cells at day 4. To definitively proof this hypothesis, we would need to do live cell analysis, which is beyond the scope of this work. The number of differentiating cells are still increasing.

*Another problem is with the “genetic interaction” between fltp and celsr1. Genetic interaction is typically shown between heterozygotes that have no phenotype on their own, or between one het with no phenotype and a homozygote with a weak phenotype. The authors make double homozygous mutants and show an enhanced phenotype. Yet each homozygote has a phenotype of its own, and therefore the result could simply be an additive phenotype. The result therefore DOES NOT show a genetic interaction and provides NO EVIDENCE that Fltp is in the PCP pathway. The result is equally consistent with distinct pathways converging on a shared output*.

We totally agree with the comment that Celsr1+Fltp could also act in two parallel pathways and we added this to the manuscript.

*Numerous other statements describing Fltp function in PCP signaling are also misleading. For example, describing Dvl2 function at BBs as a PCP function is not valid. Indeed, the MCC phenotype is not a PCP phenotype at all, and the hair cell phenotype impacts PCP, but is presumably quite downstream. Taken together, it is not reasonable to describe the results as being effectors of PCP. This is another big example of interpretation not correctly reflecting the findings*.

We agree and have toned down the statements as outlined above for reviewer 1.

Reviewer #3:

*1) Many references are made to “Detailed” subcellular co-localization studies. But these data are not at all clear in their current form. In particular the data in*
Figure 3
*are over-interpreted. It is stated that Fltp “specifically” colocalizes with newly synthesized MT plus-ends, but*
Figure 3
*shows much green where there is no red and much of the converse as well*.

We agree and apologise for the bad quality of Figure 4. We changed several panels and added a co-staining of Fltp and EB-1.

*2) The claim is made in*
Figure 3
*that fltp is localized to the apical but not the sub-apical actin in multiciliated cells, yet*
Figure 3
*shows only a single x/y image and*
Figure 3
*shows Zo1, but not actin. How is this claim justifiable from these data?*

We replaced Figure 4 and show now a Z stack which shows that Fltp is not localized to the subapical actin network (Figure 4), but to the apical actin cytoskeleton (Figure 4), cilia (Figure 4), MT plus ends (Figure 4) and around the pericentriolar matrix (Figure 4).

*3) The authors need to comment on the possible role for fltp in basal body amplification. According to*
Figure 5*, the most obvious phenotype in the mutants may not be BB docking, but the reduction in BB numbers. This issue is a big one because recent work on cyclin O from Kinter and Omran shows that BB docking and amplification may be linked*.

Thank you for the valuable comment. This is an interesting point but the phenotype observed is a clear BB docking phenotype. We reanalysed the data and have not noticed cells with significant less centrosomes in Fltp KO ALI cultures compared to WT ALIs (also compare Figure 4).